# A neural network radiative transfer model approach applied to TROPOMI's aerosol height algorithm

Swadhin Nanda[1,2], Martin de Graaf[1], J. Pepijn Veefkind[1,2], Mark ter Linden[3], Maarten Sneep[1], Johan de Haan[1], and Pieternel F. Levelt[1,2]

[1]Royal Netherlands Meteorological Institute (KNMI), Utrechtseweg 297, 3731 GA De Bilt, The Netherlands
[2]Delft university of Technology (TU Delft), Mekelweg 2, 2628 CD Delft, The Netherlands
[3]S&T Corp, Delft, The Netherlands

*Correspondence to:* Martin de Graaf (martin.de.graaf@knmi.nl)

**Abstract.** To retrieve aerosol properties from satellite measurements of the oxygen A-band in the near infrared, a line-by-line radiative transfer model implementation requires a large number of calculations. These calculations severely restrict a retrieval algorithm's operational capability as it can take several minutes to retrieve aerosol layer height for a single ground pixel. This paper proposes a forward modeling approach using artificial neural networks to speed up the retrieval algorithm. The forward model outputs are trained into a set of neural network models to completely replace line-by-line calculations in the operational processor. Results of comparing the forward model to the neural network alternative show encouraging results with good agreements between the two when applied to retrieval scenarios using both synthetic and real measured spectra from TROPOMI (TROPOspheric Monitoring Instrument) on board the ESA Sentinel-5 Precursor mission. With an enhancement of the computational speed by three orders of magnitude, TROPOMI's operational aerosol layer height processor is now able to retrieve aerosol layer heights well within operational capacity.

## 1 Introduction

Launched in October 13, 2017, The TROPOsperic Monitoring Instrument (Veefkind et al., 2012) on board the Sentinel-5 Precursor mission is the first of the satellite-based atmospheric composition monitoring instruments in the Sentinel mission of the European Space Agency. The aerosol layer height (ALH) retrieval algorithm (Sanders and de Haan, 2013; Sanders et al., 2015; Nanda et al., 2018a, b) is a part of TROPOMI's operational product suite, expected to be delivered near real time. The ALH (symbolised as $z_{\mathrm{aer}}$) retrieval algorithm, operating within the near infrared region in the oxygen A-band between 758 nm - 770 nm, exploits information about heights of scattering layers derived from absorption of photons by molecular oxygen — the amount of absorption indicates whether the scattering layer is closer or farther from the surface; if the number of photons absorbed by oxygen is higher, it suggests a longer photon path length due to an aerosol layer present closer to the surface. This principle has been applied to cloud height algorithms such as FRESCO (Fast Retrieval Scheme for Clouds from the Oxygen A-band) by Wang et al. (2008), which use look up tables for generating top of atmosphere (TOA) reflectances to compute cloud parameters. Since clouds are such efficient scatterers of light, FRESCO can approximate scattering by cloud using a Lambertian model — this simplification works for optically thick cloud layers quite well. For aerosol layers, however, such calculations

need to be done in much greater detail due to their weaker scattering properties. TROPOMI's ALH algorithm employs the science code Disamar (Determining Instrument Specifications and Methods for Atmospheric Retrievals) that uses the Layer-Based Orders of Scattering (LABOS) radiative transfer model based on the doubling-adding method (de Haan et al., 1987) that calculates reflectances at the TOA and its derivatives with respect to aerosol layer height and aerosol optical thickness ($\tau$).

These calculations are done line-by-line, requiring calculations at 3980 wavelengths to generate these TOA reflectances within the oxygen A-band. Having computed the TOA reflectance spectra, aerosol layer heights are retrieved with Optimal Estimation (OE), an iterative retrieval scheme developed by Rodgers (2000) that incorporates a priori knowledge of retrieval parameters into their estimation. Such a retrieval scheme also provides a posteriori error estimations, which are important for assimilation models and diagnosing the retrieval results.

The ALH retrieval algorithm is computationally expensive, requiring several minutes to compute $z_{aer}$ for a single ground pixel (Sanders et al., 2015). As near-real time processors need to consistently go through large volumes of data recorded by the satellite for the mission lifetime, the operational computation capability is much restricted for TROPOMI recording approximately 1.4 million pixels within a single orbit where, on average, 50,000 pixels are typically identified as aerosol contaminated pixels (with a UVAI value greater than 0.0) for retrieving aerosol layer height. This places a steep requirement

on the computational infrastructure to process all possible pixels from a single orbit. The online radiative transfer model severely limits the ALH data product, processing only a small fraction of the total possible pixels within a single orbit while compromising the timeliness of the data delivery.

The bottleneck identified here is the large number of calculations that the forward model has to compute to retrieve information on weak scatterers such as aerosols. Several steps to circumvent this bottleneck exist, such as using correlated

k-distribution method to reduce the number of calculations (Hasekamp and Butz, 2008), using a look up table for calculating forward model outputs, or entirely foregoing the forward model and directly retrieving $z_{aer}$ from observed spectra using neural networks (Chimot et al., 2017, 2018). Studies by Sanders and de Haan (2016) have shown that the look up table for reflectance alone measure up to 46 GB in size, and perhaps similar or larger sizes for the derivatives. Chimot et al. (2017) describe an approach using a radiative transfer model to generate slant column densities of the $O_2$-$O_2$ band at 477 nm from Ozone Mon-

itoring Instrument (OMI) measurements for different aerosol optical depths (among other input parameters) to train several artificial neural network models that directly retrieve aerosol layer height. Operationally, their neural network models use the MODIS aerosol optical depth at 550 nm and retrieved OMI slant column densities, thereby entirely foregoing line-by-line calculations and significantly speeding up the retrieval algorithm. The trained neural network models directly retrieved aerosol layer heights from spectra measured by OMI on board the NASA Aura mission, without using line-by-line calculations or an

iterative estimation step such as OE (Chimot et al., 2018). A similar example of retrievals is the ROCINN (Retrieval of Cloud Information using Neural Networks) cloud algorithm developed by Loyola (2004) which uses neural networks to compute convolved reflectance spectra to retrieve cloud properties. These retrievals show the exploitable capabilities of artificial neural networks in the context of retrieving atmospheric properties from oxygen absorption bands.

The work of Chimot et al. (2018) and Loyola et al. (2018) bring to light the efficacy of artificial neural networks in satel-

lite remote sensing of oxygen absorption bands for retrieving properties of scattering species in the atmosphere. This paper

discusses a method inspired by Chimot et al. (2017) and Loyola (2004) to retrieve aerosol layer height from oxygen A-band measurements by TROPOMI. While Chimot et al. (2017) directly retrieve aerosol layer heights from their neural network models, the operational algorithm in this paper utilises neural networks to calculate top-of-atmosphere radiances in the forward model. This is subsequently used by an optimal estimation scheme to retrieve aerosol layer heights. Similarly while Loyola (2004) derive top-of-atmosphere sun-normalised radiances only for their cloud property retrieval algorithm, the method in this paper has dedicated neural network models that calculate the Jacobian as well as the top-of-atmosphere sun-normalised radiances. By reducing the time consumed for calculating forward model outputs, computational efficiency of TROPOMI's aerosol layer height retrieval algorithm can be significantly improved.

Section 2 introduces the operational aerosol layer height algorithm and discusses the line-by-line forward model. The neural network forward model approach is detailed in section 3, and its verification on a test data set is discussed in same section. This approach is then applied to various test cases using synthetic and real TROPOMI spectra (section 4) before concluding in section 5.

## 2 The TROPOMI aerosol layer height retrieval algorithm

The TROPOMI aerosol layer height is one of the many algorithms that exploit vertical information of scattering aerosol species in the oxygen A-band (Timofeyev et al., 1995; Gabella et al., 1999; Corradini and Cervino, 2006; Pelletier et al., 2008; Dubuisson et al., 2009; Frankenberg et al., 2012; Sanghavi et al., 2012; Wang et al., 2012; Sanders and de Haan, 2013; Hollstein and Fischer, 2014; Sanders et al., 2015; Geddes and Bösch, 2015; Sanders and de Haan, 2016; Colosimo et al., 2016; Davis et al., 2017; Xu et al., 2017; Nanda et al., 2018b; Zeng et al., 2018). These methods invert a forward model that describes the atmosphere, to compute the height of the scattering layer. This section discusses the setup of the TROPOMI ALH retrieval algorithm, which consists of the inversion of a forward model representing the atmosphere using optimal estimation as the retrieval method, and a description of the forward model.

### 2.1 The retrieval method

The cost function $\chi^2$ represents the departure of the modeled reflectance $\boldsymbol{F}(\boldsymbol{x})$ from the observed reflectance $\boldsymbol{y}$ constrained with by the measurement error covariance matrix $\mathbf{S}_\epsilon$, and is defined as

$$\chi^2 = [\boldsymbol{y} - \boldsymbol{F}(\boldsymbol{x})]^T \mathbf{S}_\epsilon^{-1} [\boldsymbol{y} - \boldsymbol{F}(\boldsymbol{x})] + (\boldsymbol{x} - \boldsymbol{x_a})^T \mathbf{S_a}^{-1} (\boldsymbol{x} - \boldsymbol{x_a}). \tag{1}$$

Minimising this cost function for a particular $z_{\mathrm{aer}}$ and $\tau$ (the elements of the state vector $\boldsymbol{x}$ to be retrieved and fitted) gives us the final retrieval product. Minimising this cost function for a particular $z_{\mathrm{aer}}$ and $\tau$ (the elements of the state vector $\boldsymbol{x}$ to be retrieved and fitted) gives us the final retrieval product. This definition of the cost function is unique to OE, as it is constrained with a priori knowledge of the state vector $\boldsymbol{x}$ (represented by $\boldsymbol{x_a}$) and the a priori error covariance matrix $\mathbf{S_a}$. In the TROPOMI ALH processor's OE framework, the a priori state vector is fixed at specific values, usually 200 hPa above the surface for $z_{\mathrm{aer}}$

and 1.0 for $\tau$ at 760 nm. The a priori error of the $z_{\mathrm{aer}}$ is fixed at 500 hPa, and the same for $\tau$ is 1.0, to allow freedom for the variables in the estimation (this also reduces the impact of the a priori on the retrieval). The forward model is employed to simulate the measured reflectance spectrum with model parameter $\boldsymbol{x}$ with

$$F(\mathbf{x})(\lambda) = \frac{\pi I(\lambda)}{\mu_0 E_0(\lambda)}, \tag{2}$$

where $I$ and $E_0$ represent the Earth radiance and solar irradiance, respectively, with the cosine of the solar zenith angle ($\theta_0$) denoted by $\mu_0$. Since the forward model is non-linear, a Gauss-Newton iteration is employed to the updated state vector as following,

$$\boldsymbol{x_{i+1}} = \boldsymbol{x_a} + [\mathbf{K_i}^T \mathbf{S}_\epsilon^{-1} \mathbf{K_i} + \mathbf{S_a}^{-1}]^{-1} \mathbf{K_i}^{-1} \mathbf{S}_\epsilon^{-1} [\boldsymbol{y} - \boldsymbol{F}(\boldsymbol{x}) + \mathbf{K_i}(\boldsymbol{x_i} - \boldsymbol{x_a})], \tag{3}$$

where $\mathbf{K_i}$ is the matrix of derivatives (Jacobian) of the reflectance with respect to state vector parameters at the current iteration $i$. The derivatives are calculated semi-analytically similar to the method described by Landgraf et al. (2001). $n^{\mathrm{th}}$ iterative estimate is convergent to a solution if the relative changes in the state vector is less than the expected precision (usually fixed at a certain value). The retrieval is decided to be failed if the number of iterations exceeds the maximum number of iterations (usually set at 12), or if the state vector parameters are projected outside the respective boundary conditions. Retrieval errors are derived from the a posteriori error covariance matrix $\hat{\mathbf{S}}$, computed as

$$\hat{\mathbf{S}} = [\mathbf{K}^T \mathbf{S}_\epsilon^{-1} \mathbf{K} + \mathbf{S_a}^{-1}]^{-1}. \tag{4}$$

## 2.2 The Disamar forward model and its many simplifications of atmospheric properties

Optimal estimation iteratively simulates TOA radiance spectra until the convergence of $\chi^2$ (Equation 1). For this, disamar computes reflectances at a high resolution wavelength grid. The computed high resolution reflectances are combined with a reference solar spectrum derived from Chance and Kurucz (2010) to obtain a high resolution Earth radiance. The high resolution Earth radiance and the solar spectrum are convolved with the instrument spectral response function to obtain Earth radiance and solar irradiance spectrum in the instrument's wavelength grid, before finally computing the reflectance spectrum in the instrument grid using Equation 2. It is important to note that the steps of including the reference solar spectrum to compute reflectances in the instrument's wavelength grid are not undertaken by the neural network algorithm. The neural network aerosol layer height retrieval algorithm directly convolves the reflectance. The difference between including an excluding a reference spectrum in the convolution process results in differences in the order of 4% to 5% around 762 nm and 766 nm. Further on in this paper, a direct comparison between disamar retrievals of aerosol layer height and retrievals with the neural network algorithm is provided.

Reflectances are calculated by accounting for scattering and absorption of photons from their interactions with aerosols, the surface, and molecular species. Molecular scattering of photons in the oxygen A-band is described by Rayleigh scattering, and

absorption is described by photon-induced magnetic dipole transition between $b^1\Sigma_g^+ \leftarrow X^3\Sigma_g^- (0,0)$ electric potential levels of molecular oxygen, and collision-induced absorption between $O_2$-$O_2$ and $O_2$-$N_2$. The total influence of the $O_2$ A-band on the TOA reflectance is described by its extinction cross-section, which is a sum of the three aforementioned contributions. As the vertical distribution of oxygen is exactly known, the extinction cross-section can be exploited to retrieve $z_{aer}$ from satellite measurements of the oxygen A-band. For this, Disamar calculates absorption (or extinction) cross sections at 3980 wavelengths within the range 758 nm - 770 nm.

To reduce the number of calculations, various atmospheric properties are simplified. As the Rayleigh optical thickness is low at 760 nm, Disamar only computes the monochromatic component of light by calculating the first element of the Stoke's vector. The exclusion of higher order Stoke's vector elements of the radiation fields has not shown to be a significant source of error (Sanders and de Haan, 2016).

Calculating the influence of Rotational Raman Scattering (RRS) is also ignored, as it is a computationally expensive step. While this exclusion of RRS is not advised by literature (Vasilkov et al., 2013; Sioris and Evans, 2000), preliminary experiments by Sanders and de Haan (2016) have ascertained that the errors in the retrieved aerosol layer height resulting from ignoring RRS of the oxygen A-band in the forward model are significantly smaller than the effect of other model errors such as errors due to incorrect surface albedo. Therefore, RRS has been historically not simulated in the forward model of the KNMI aerosol layer height retrieval algorithm. The atmosphere is assumed cloud-free, which is a required simplification as the retrieval of $z_{aer}$ in the presence of clouds is still challenging (Sanders et al., 2015) and thereby is performed only for pixels which are unlikely to contain clouds. Compared to totally cloud-free scenes, errors in retrieved $z_{aer}$ are large for cloud-free scenes containing undetected optically thin cirrus clouds (Sanders et al., 2015). The fraction of the pixel containing aerosols is assumed to be 100%, which further simplifies the representation of aerosols within the atmosphere.

Perhaps the largest simplification of the atmosphere lies in model's description of aerosols, assumed to be distributed in a homogeneous layer at a height $z_{aer}$ with a 50 hPa thickness, a fixed aerosol optical thickness ($\tau$) and a single scattering albedo ($\omega$) of 0.95 (so, scattering aerosols). A Henyey-Greenstein model (Henyey and Greenstein, 1941) with an asymmetry parameter $g$ value of 0.7 is used to parameterize the aerosol scattering phase function, which is one of the widely used approximations. These fixed aerosol optical properties have been derived from AERONET data and tested by Sanders et al. (2015), who retrieved $z_{aer}$ from GOME-2 spectra to show that the algorithm is robust to fixing aerosol model parameters such as the single scattering albedo and the Henyey-Greenstein phase function asymmetry parameter. The surface is assumed to be an isotropic reflector with a brightness described by its Lambertian Equivalent Reflectivity (LER). This is also an important simplification, requiring less computations over other surface models such as a Bi-directional Reflectance Model. Although the forward model is capable of including sun-induced chlorophyll fluorescence into the retrieval, it is currently being considered for a future implementation of TROPOMI's operational ALH retrieval algorithm. Lastly, the atmosphere is spherically corrected for incoming solar radiation and remains plane-parallel for outgoing Earth radiance.

These simplifications in the Disamar forward model are a necessity for the line-by-line aerosol layer height algorithm, owing to its slow computational speed. The speed up of forward model simulation encourages increasing the complexity of simulation assumption.

## 2.3 Application to TROPOMI

TROPOMI's near infrared (NIR) spectrometer records data between 675 nm - 775 nm, spread across two bands — band 5 contains the oxygen B-band and band 6 the oxygen A-band. The spectral resolution, which is described by the full width at half maximum (FWHM) of the instrument spectral response function (ISRF), is 0.38 nm with a spectral sampling interval of 0.12 nm. The spatial resolution is around 7 km × 3.5 km for band 5 and 6. Initial observations from the TROPOMI NIR spectrometer show a signal to noise ratio (SNR) of 3000 in the continuum before the oxygen A-band. The instrument polarization sensitivity is reduced to below 0.5% by adopting the technology of the polarization scrambler of OMI (Veefkind et al., 2012; Levelt et al., 2006). Disamar utilizes TROPOMI's swath-dependent ISRFs to convolve $I(\lambda)$ and $E_0(\lambda)$ into $I(\lambda_i)$ and $E_0(\lambda_i)$ in the instrument's spectral wavelength grid, after which the modeled reflectance is calculated using Equation 2.

Input parameters required by the TROPOMI ALH retrieval algorithm encompass satellite observations of the radiance and the irradiance, solar-satellite geometry, and a host of atmospheric and surface parameters required for modeling the interactions of photons within the Earth's atmosphere (see Table 1). Meteorological parameters are taken from ECMWF (European Centre for Medium-range Weather Forecast), including the temperature-pressure profile at 91 atmospheric levels (of which the surface is a part). The various geophysical parameters are interpolated to TROPOMI's ground pixels using nearest neighbour interpolation.

TROPOMI incorporates information from the VIIRS instrument to detect the presence of cirrus clouds in the measured scene (using a cirrus reflectance threshold of 0.01). This information is further combined with cloud fraction retrievals by the TROPOMI FRESCO algorithm (maximum cloud fraction of 0.6), and the difference between the scene albedo in the database in the UV band and the apparent scene albedo at the same wavelength calculated using a lookup table (if the difference is larger than 0.2, it suggests cloud contamination). A combination of these different cloud detection strategies results in the cloud_warning flag in the level-2 TROPOMI ALH product. In this paper, however, we use a strict FRESCO cloud fraction filter of 0.2 alone to remove cloudy pixels.

**Table 1.** Input parameters required for retrieving aerosol layer height using TROPOMI measured spectra.

| Parameter | Source | Remarks |
| --- | --- | --- |
| Radiance and irradiance | TROPOMI Level-1b product | |
| SNR measured spectrum | TROPOMI Level-1b product | |
| Geolocation parameters | TROPOMI Level-1b product | |
| Surface albedo | GOME-2 LER database | Tilstra et al. (2017) |
| Meteorological parameters | ECMWF | 17km horizontal resolution |
| Cloud fraction | TROPOMI Level-2 FRESCO product | |
| Absorbing aerosol index (AAI) | TROPOMI Level-2 AAI product | |
| Land-sea mask | NASA Toolkit | |
| Surface altitude | GMTED 2010 | pre-averaged |

Calculation of TOA reflectance and its derivatives with respect to $z_{\mathrm{aer}}$, and $\tau$ in a line-by-line fashion takes approximately 40-60 seconds to complete on a computer equipped with Intel(R) Xeon(R) CPU E3-1275 v5 at a clock speed of 3.60 GHz. In an iterative framework such as the Gauss-Newton method, the retrieval of $z_{\mathrm{aer}}$ can take between 3-6 iterations depending on the amount of aerosol information available in the observed spectra, requiring several minutes to compute retrieval outputs for a specific scene. If these retrievals fail by not converging within the maximum number of iterations, the processor can waste up to 10 minutes on a pixel without retrieving a product. In order to compute Disamar's outputs quicker, a neural network implementation is discussed in the next section.

## 3 The neural network (NN) forward model

Artificial neural networks consist of connected processing units, each individually producing an output value given a certain input value. The interaction of these individual processing units, also known as nodes (or neurons), enable the connecting network to map a set of inputs (also known as the input layer) to a set of outputs (or, the output layer). The connections are known as weights whose value symbolises the strength of a connection between two nodes. Since the nodes connect inputs to the outputs, higher values in a set of connecting weights represent a stronger influence of a particular parameter in the input layer over a particular parameter in the output layer. These weights are determined after training the neural network.

The training (or optimisation) of a neural network begins with a training data set containing many instances of input and output layer elements. As true values of the output layer for a given set of inputs are exactly known in the training data set, the biased output of the neural network calculated after using randomised, non-optimised weights can be easily calculated. These biases are called prediction errors, an essential element in the optimization of the neural network weights. The mean squared error (MSE) between the true output and the calculated output is also called the loss function (henceforth annotated as $\Delta$), which is synonymous to a cost function (Equation 5),

$$\Delta = \frac{1}{n_\lambda} \sum_{\forall \lambda} (nn_\lambda - o_\lambda)^2 \tag{5}$$

where $\lambda$ is the wavelength, $n_\lambda$ represents the number of elements in the output layer, $nn_\lambda$ represents the calculated output for wavelength via forward propagation, and $o_\lambda$ are the outputs in the training data set. The weights are updated using optimisers such as the ADAM optimiser (Adaptive Moment Estimation) by Kingma and Ba (2014) to minimise $\Delta$, within set number of iterations.

### 3.1 The TROPOMI NN forward model for the ALH retrieval algorithm

The standard architecture of the NN-augmented operational aerosol layer height processor includes three neural network models for estimating top of atmosphere sun-normalised radiance, the derivative of the reflectance with respect to $z_{\mathrm{aer}}$, and the same for $\tau$. It is also possible to assign the neural network to compute the reflectance instead of the sun-normalized radiance — the results will not change. The definition of sun-normalised radiance used in this paper is the ratio of Earth radiance to

solar irradiance. Disamar calculates derivatives with respect to reflectance, which is the sun-normalised radiance multiplied by the ratio of $\pi$ and cosine of solar zenith angle. All three neural network models share the same input model parameters. Optimising a single neural network model for all three forward model outputs is not necessary; the correlations between the input parameters and the different forward model outputs are different, which can complicate the optimisation of a general-purpose neural network. This paper, however, acknowledges modern developments in neural network optimisation techniques that now afford selectively optimising a neural network for different tasks (Kirkpatrick et al., 2016; Wen and Itti, 2018).

The models are trained using the python Tensorflow module (Abadi et al., 2015), and further implemented into an operational processor using C++ interface to Tensorflow. These neural network models require training data containing Disamar input and output parameters and a connecting architecture that encompasses the input feature vector containing scene-varying model parameters, the number of hidden layers, number of nodes in each hidden layer, and an activation function that maps the input to the final output layer containing Disamar outputs. In Tensorflow, the derivative of $\Delta$ with respect to the weights are computed using reverse-mode automatic differentiation, which computes numerical values of derivatives without the use of analytical expressions (Wengert, 1964).

The inputs for NN are referred together as the feature vector. The choice of the parameters included into the feature vector is a very important factor deciding the performance of the neural network. The primary classes of model parameters (relevant to retrieving $z_{\mathrm{aer}}$) varying from scene to scene are solar-satellite geometry, aerosol parameters, meteorological parameters and surface parameters (Table 2). The various aerosol parameters that are fixed from scene to scene are the aerosol single scattering albedo ($\omega$), the asymmetry factor of the phase function, and the angstrom exponent, as they are also fixed in the line-by-line operational aerosol layer height processor. The scattering phase function of aerosols is currently limited to a Henyey-Greenstein model with a fixed $g$ value of 0.7 to mimic Disamar. Surface pressure as well as the temperature-pressure profile are two important meteorological parameters relevant to retrieving $z_{\mathrm{aer}}$. A difference between Disamar and NN models is the definition of this temperature information in the input. Disamar requires the entire temperature-pressure profile of the atmosphere, whereas NN only uses the temperature at $z_{\mathrm{aer}}$. Surface albedo is specified at 758 nm as well as 772 nm in Disamar, whereas it is only specified at 758 nm in the feature vector of NN. In general there is a greater scope to add detailed information in Disamar. However, Disamar has historically incorporated many simplifications in order to reduce computational time. The current NN model is developed with the aim to mimic Disamar as much as possible, without including additional state vector elements into the retrieval, such as chlorophyll fluorescence, aerosol optical properties, cloud properties, and so on.

## 3.2 Training the neural networks

Since the NN forward model is specifically designed for TROPOMI, the solar-satellite geometry is selected to represent TROPOMI orbits for the training data. Meteorological parameters for the locations associated with these solar-satellite geometries are derived from the 2017 60-layer ERA-Interim Reanalysis data (Dee et al., 2011), and aerosol and surface parameters are randomly generated within their physical boundaries. This training data generation strategy spans the entire set of TROPOMI solar and viewing angles as well as meteorological parameters.

**Table 2.** Scene-dependent input model parameters for the NN model. See also Figure 1 for a histogram of the input parameters. The solar-satellite geometry parameters are generated in combinations conforming to the ones encountered by TROPOMI's orbits.

| Parameter class | Model Parameters | Remarks | limits |
|---|---|---|---|
| Geometry | Solar zenith angle ($\theta_0$) | in feature vector | $8.20°$ - $80.0°$ |
| | Viewing zenith angle ($\theta$) | in feature vector | $0.0°$ - $66.60°$ |
| | Solar azimuth angle ($\phi_0$) | in feature vector | $-180.0°$ - $180.0°$ |
| | Viewing azimuth angle ($\phi$) | in feature vector | $-180.0°$ - $180.0°$ |
| Aerosol parameters | Aerosol pixel fraction | fixed | 1.0 |
| | Single scattering albedo ($\omega$) | fixed | 0.95 |
| | Aerosol optical thickness ($\tau$) | in feature vector | 0.05 - 5.0 |
| | Aerosol layer height ($z_{\text{aer}}$) | in feature vector | 75 hPa - 1000.0 hPa |
| | Aerosol layer thickness ($p_{\text{thick}}$) | varied but excluded from feature vector | 50 hPa - 200 hPa |
| | Scattering phase function | fixed | Henyey-Greenstein |
| | asymmetry factor ($g$) | fixed | 0.7 |
| | Angstrom exponent (Å) | fixed | 0.0 |
| Meteorological parameters | Temperature | in feature vector | temperature at $z_{\text{aer}}$ |
| Surface parameters | Surface pressure ($p_s$) | in feature vector | 520 hPa - 1048.50 hPa |
| | Surface reflectance model | LER | |
| | Surface albedo ($A_s$) | in feature vector | 2.08E-7 - 0.70 |

Generally, the required training data size increases with increasing non-linearity between input and output layers in a neural network — there is no specific method to accurately determine the required sample size before training. The number of spectra generated for the training set was determined by training different models with different number of spectra in the training set ranging from 1,000 to 600,000. In general it was observed that incorporating more data resulted in a better neural network model. In order to test the trained neural network model, a choice of 500,000 spectra were selected. Finding the most optimal neural network configuration requires testing the trained neural network model. To that extent, the training data set was split into a training-testing split, where the model was trained on a majority of the training data set and tested on the remaining minority. Once trained, the model was tested again on a test data set with 100,000 scenes outside of the training data set. These spectra were generated using Disamar with model parameter ranges described in Table 2. Figure 1 plots the distribution of the input parameters necessary for training the neural network. The neural network model accepts solar azimuth and viewing azimuth angles separately, however they are plotted together as relative azimuth angle in Figure 1 to save space. The generation of this training data set is by far the most time consuming step since each Disamar run requires between 50-60 seconds to generate the synthetic spectra. Once the data has been generated, it is prepared for training the neural network models in NN. This is done by data normalisation, achieved by subtracting the mean of each of the training input and output parameters and dividing the difference by its standard deviation, which makes the learning process quicker by reducing the search space for the optimizer. The offset and scaling parameters are important, as the neural network computes outputs within this scaled range, which needs

to be re-scaled back to physical values. This training requires a few hours on an Intel(R) Xeon(R) CPU E3-1275 v5 at a clock speed of 3.60 GHz.

The most optimal configurations for each of the three NN models are determined by the number of hidden layers, the number of nodes on each layer and the chosen activation function for which the discrepancy between the modeled output for specific inputs and the truth (derived from Disamar) is minimal. The difference between the outputs calculated by Disamar and NN for these three models provide insight on their performance.

In order to test the most optimal number of layers, the most optimal number of nodes per each layer and the activation function, several neural network configurations were trained for 250,000 iterations and their summed losses (defined as $\Delta \times n_\lambda$) were compared to find out which was the best configuration. Figure 2 plots the summed losses as a function of training iteration for different configurations.

To begin, with 50 nodes per each hidden layer, three neural networks for each of the three models were trained — one-layered, two-layered and three-layered. The neural network models performed best with at least two hidden layers (Figure 2a). For all three models, their two-layered versions show a similar summed loss to their three-layered alternatives, with the summed loss for the two-layered $\text{NN}_{\text{disamar}}(\text{K}_\tau)$ showing more stability with training epoch. Therefore, a simpler two-layered architecture is chosen for all three models. Continuing on, three other architectures for each of the three models were chosen with 50, 100, and 200 nodes for each of the two hidden layers. The results that with more training steps, the choice of 100 nodes for each of the two layers has a compromise between summed training loss and simplicity (Figure 2b), especially for $\text{NN}_{\text{disamar}}(\text{K}_\tau)$. Finally, going ahead with a two-layered and 100 nodes for each layer configuration, three activation functions namely the sigmoid function, the hyperbolic tangent function (tanh) and the rectified linear unit (relu) function were tested for each of the neural network models (Figure 2c). In this case, while all functions converge to similar summed loss values by 250,000 iterations, the sigmoid function has a good compromise between training loss and stability. Figure 3 gives a graphic representation of the neural network model.

The finalised configurations were then trained for one million iterations after which they were applied to the test data set to study prediction errors. Figure 4 plots the performance of each of the neural networks trained on the testing data set. An error analysis revealed that the trained neural networks were capable of calculating Disamar outputs with low errors, generally within 1-3% of Disamar calculations. Averaged convolved errors of the neural network model for the sun normalised radiance ($\text{NN}_I$) did not exceed 1%. The neural network model for the derivative of the reflectance with respect to $\tau$ and $z_{\text{aer}}$ perform well in general for parts of the spectrum with large oxygen absorption cross sections, where the value of the derivatives are high (indicating a higher amount of information content from those specific wavelength regions). Errors in the deepest part of the R-branch between 759 nm and 762 nm and the P-branch between 752.50 nm and 765 nm, do not exceed 3% for $\text{NN}_{\text{K}_{\text{zaer}}}$. The same can be said for $\text{NN}_{\text{K}_\tau}$, which displays errors in the range of 1% in the same wavelength region. For wavelengths outside of the deepest parts of the R and P-branch, the relative errors are large, and exceed 10% easily. However, the relative errors are calculated as the absolute value of the difference between the true spectrum and the neural network calculated spectrum, divided by the true spectrum. These values can be very large when the value of the true spectrum is very small, which is the

case for the derivatives outside the deepest part of the R and P branches. The consequence of these errors in a retrieval scenario from synthetic and real spectra are discussed in the following section.

## 4    Comparison between Disamar and NN aerosol layer height retrieval algorithms

To test the NN augmented retrieval algorithm, we apply the generated NN models to synthetic test data and real data from
TROPOMI, and compare its retrieval capabilities to those of Disamar. The synthetic data were produced using the Disamar radiative transfer model because of which we expect the online radiative transfer retrievals to be generally better than the NN-based retrievals. The aerosol model used in the retrieval is as in Section 2.2, using fixed parameters for aerosol single scattering albedo, aerosol layer thickness and aerosol scattering phase function.

### 4.1    Performance of NN versus Disamar in retrieving aerosol layer height in the presence of model errors

A comparison of biases (in the presence of model errors) in the final retrieved solution is indicative of the efficacy of NN in
replacing Disamar to retrieve ALH. To directly compare $z_{\mathrm{aer}}$ retrieval capabilities of Disamar and NN, radiance and irradiance spectra convolved with a TROPOMI slit function were generated to replicate TROPOMI-measured spectra. Bias is defined as the difference between retrieved and true aerosol layer height (i.e., retrieved - true). A total of 2000 scenes for four synthetic experiments were generated from the test data set containing TROPOMI geometries, with randomly varied model errors in aerosol single scattering albedo, Henyey-Greenstein phase function asymmetery parameter, and surface albedo (described in
Table 3). Figure 5 compares the retrieved $z_{\mathrm{aer}}$ from line-by-line and neural network approaches for each of the synthetic experiments. A histogram of these differences is plotted in Figure 6.

The retrieved aerosol layer heights from Disamar and NN in the presence of model errors in aerosol layer thickness were found to be almost similar (Figure 5a), with a Pearson correlation coefficient close to 1.0. Introducing model errors in other
aerosol properties such as single scattering albedo (Figure 5b) and scattering phase function (Figure 5c) also resulted in a similar agreement between Disamar and NN retrieved aerosol layer heights. Furthermore, both methods retrieved similar aerosol layer heights in the presence of model errors in surface albedo as well (Figure 5d).

A total of 5558 retrievals out of the 8000 difference cases converged to a final solution. On average, $z_{\mathrm{aer}}$ retrieved using NN differed by approximately 5.0 hPa from the same using Disamar (Figure 6), with a median of approximately 2.0 hPa. The
spread of the retrieval differences were minimal, with a majority of the retrievals differing by less than 13.0 hPa. Differences close to and above 100.0 hPa did exist, but such retrievals were very uncommon.

Out of the 8000 scenes within the synthetic experiment, NN retrieved aerosol layer heights for 546 scenes where Disamar did not. Contrariwise, 586 scenes converged for Disamar and not for NN. A comparison of the biases from these odd retrieval results is plotted in Figure 7, which indicates that retrievals from NN in cases where Disamar fails are realistic as the distribution
of the biases is very similar to those cases when Disamar succeeds and NN does not (Figure 7). Retrievals using the NN forward model on average required three more iterations to reach a solution when compared to the same by Disamar. Similarly, retrievals from Disamar had a significantly lower minimised cost function (less by four orders of magnitude on average) at the end of

the retrieval when compared to NN. This is within expectation as NN cannot truly replicate Disamar. Having tested the NN augmented retrieval algorithm in a synthetic environment, the retrieval algorithm was installed into the operational TROPOMI processor for testing with real data.

**Table 3.** A count of converged and non-converged results from synthetic experiments comparing retrieved aerosol layer heights between Disamar and NN.

| | experiment | | Disamar | | NN | |
|---|---|---|---|---|---|---|
| model parameter | value in sim | value in ret | converged | non converged | converged | non converged |
| $p_{thick}$ | 200 hPa | 50 ha | 1641 | 359 | 1550 | 450 |
| $\omega$ | 0.93 - 0.96 | 0.95 | 1396 | 604 | 1412 | 588 |
| $g$ | 0.67 - 0.73 | 0.7 | 1571 | 429 | 1567 | 433 |
| $A_s$ | $0.95A_s$ - $1.05A_s$ | $A_s$ | 1536 | 464 | 1575 | 425 |

### 4.2 Application to December 2017 Californian forest fires observed by TROPOMI

The December 2017 Southern California wildfires have been attributed to very low humidity levels, following delayed autumn precipitation and severe multi-annual drought (Nauslar et al., 2018). Particularly on December 12, the region of the fires was cloud-free, owing to high-pressure conditions. A MODIS Terra image of the plume and the retrieved AAI from TROPOMI is plotted in Figure 8. The biomass burning plume extended well beyond the coastline and over the ocean (Figure 8a), which provides a roughly cloud-free and low surface brightness test case for implementing the aerosol layer height retrieval algorithm. The AAI values were above 5.0 in the bulk of the plume (Figure 8b), indicating a very high concentration of elevated absorbing aerosols. Pixels with an AAI value less than 1.0 were excluded from the retrieval experiment. Cloud-contaminated pixels were removed from the data selected for processing using the FRESCO cloud mask product from TROPOMI (maximum cloud fraction of 0.2), but parts of the biomass burning plume that did not contain any clouds were also removed as the cloud fraction values for these pixels were higher than the threshold. This is because FRESCO-based cloud fraction values over cloud-free scenes containing aerosols (biomass burning aerosols in this case) are generally expected to be positively biased. The retrieval algorithms did not process pixels in the coastline, where the surface albedo retrieval is likely to be wrong.

Figure 9 compares the retrieved $z_{aer}$ over the plume using the line-by-line and neural network based forward models, respectively. The number of the converged retrievals is 7418 for the line-by-line algorithm, but 7370 for the neural network algorithm. The differences between $z_{aer}$ (disamar) and $z_{aer}$ (NN) go up to as much as 0.5 km (Figure 9c). A majority of the negative differences are for the part of the plume extending from the coast between 47°N and 40°N. Figure 10 provides plots for further comparison between the two retrieval techniques. The neural network augmented processor retrieved aerosol layer heights which were (on average) less than 50.0 meters apart from the same by the line-by-line counterpart (Figure 10b). The standard deviation of the differences are approximately 160 meters, which indicates the presence of outliers. However, a majority of the differences in the two retrievals are less than 100 meters; this is indicated by the 15[th] and the 85[th] percentile of

these differences of -115.0 meters and 40.0 meters respectively. Although the retrieval algorithms have good agreement, they primarily differed for the lower aerosol loading scenes (Table 4). The majority of the pixels where the neural network algorithm differed from the line-by-line counterpart by more than 200 meters were for AAI values less than 2.0 (Figure 10c). Most of these biases were caused by an over-estimation of the retrieved aerosol layer height using the neural network algorithm, in comparison to the same from disamar. Pixels with AAI values larger than 5.0 also showed a consistent bias of 60 meters with a standard deviation of 30 meters. This bias is not well understood.

**Table 4.** Statistics of difference between retrieved $z_{aer}$ from Disamar and NN from Figure 9c.

| AAI [-] | number of samples | mean [m] | median [m] | standard deviation [m] | 15th percentile [m] | 85th percentile [m] |
|---------|-------------------|----------|------------|------------------------|---------------------|---------------------|
| <2.0    | 3227              | -50.74   | -62.10     | 206.44                 | -228.65             | 108.31              |
| 2.0 - 3.0 | 2723            | -54.96   | -43.20     | 110.75                 | -184.85             | 67.10               |
| 3.0 - 5.0 | 1167            | 10.32    | 19.42      | 63.65                  | -61.63              | 65.26               |
| >5.0    | 253               | 61.35    | 61.00      | 30.954                 | 26.56               | 95.22               |

The time required by the line-by-line operational processor was $184.01\pm0.50$ seconds per pixel, whereas the same for the neural network processor was $0.167\pm 0.0003$ seconds per pixel. The neural network algorithm shows an improvement in the computational speed by three orders of magnitude over the line-by-line retrieval algorithm. The computational speed gained from implementing NN enables retrieval of aerosol layer heights from all potential scenes in the entire orbit within the stipulated operational processing time slot.

## 5   Conclusions

Of the algorithms that currently retrieve TROPOMI's suite of level-2 products, the aerosol layer height processor is an example of one that requires online radiative transfer calculations. These online calculations have traditionally been tackled with KNMI's radiative transfer code disamar, which calculates (among other parameters) sun-normalised radiances in the oxygen A-band. There are, in total, 3980 line-by-line calculations per iteration in the optimal estimation scheme, requiring several minutes to retrieve aerosol layer height estimates from a single scene. This limits the yield of the aerosol layer height processor significantly.

The bottleneck is identified to be the number of calculations Disamar needs to do at every iteration of the Gauss-Newton scheme of the estimation process. As a replacement, this paper proposes using artificial neural networks in the forward model step. Three neural networks are trained, for the sun-normalised radiance and the derivative of the reflectance with respect to aerosol layer height and aerosol optical thickness, the two state vector elements. As the goal is to replicate and replace Disamar, line-by-line forward model calculations from Disamar were used to train these neural networks. A total of 500,000 spectra were generated using Disamar, and each of the neural network models was trained for a total of 1 million iterations with the mean squared error between the training data output and the neural network output being the cost function to be minimised in the optimisation process.

Over a test data set with 100,000 different scenes unique from the training data set, the neural network models performed well, with errors not exceeding 1-3% in general in the predicted spectra and derivatives. Having tested the neural network models for prediction errors in the forward model output spectra, they were implemented into the aerosol layer height breadboard algorithm and further tested for retrieval accuracy. In order to do so, experiments with synthetic as well as real data were conducted. The synthetic scenes included 2000 spectra with different model errors in aerosol and surface properties. In these cases, the neural network algorithm showed very good compatibility with the aerosol layer height algorithm, since it was able to replicate the biases satisfactorily.

We evaluate aerosol layer heights retrieved from TROPOMI measurements over Southern California on 12 December, 2017, when the fire plume extensively floats from land to ocean over a dry and almost cloudless scene. Operational retrievals using both Disamar and the neural network forward models showed very similar results, with a few outliers around 500 meters for pixels containing low aerosol loads. These biases were outweighed by the upgrade in the computational speed of the retrieval algorithm, as the neural network augmented processor observed a speedup of three orders of magnitude, making the aerosol layer height processor operationally feasible. Having achieved this improvement in its computational performance, the aerosol layer height algorithm is planned to be operationally retrieving the product for the all possible pixels in each orbit of TROPOMI. Such a boost in processor output allows for better analyses of retrievals and opens the possibility to remove some of the forward model simplifications mentioned in Section 2.2, which paves the way for further developing the TROPOMI aerosol layer height algorithm.

*Competing interests.* The author declares no conflict of interests in the work expressed in this publication.

*Acknowledgements.* This publication contains modified Copernicus Sentinel data. This research is partly funded by the European Space Agency (ESA) within the EU Copernicus programme.

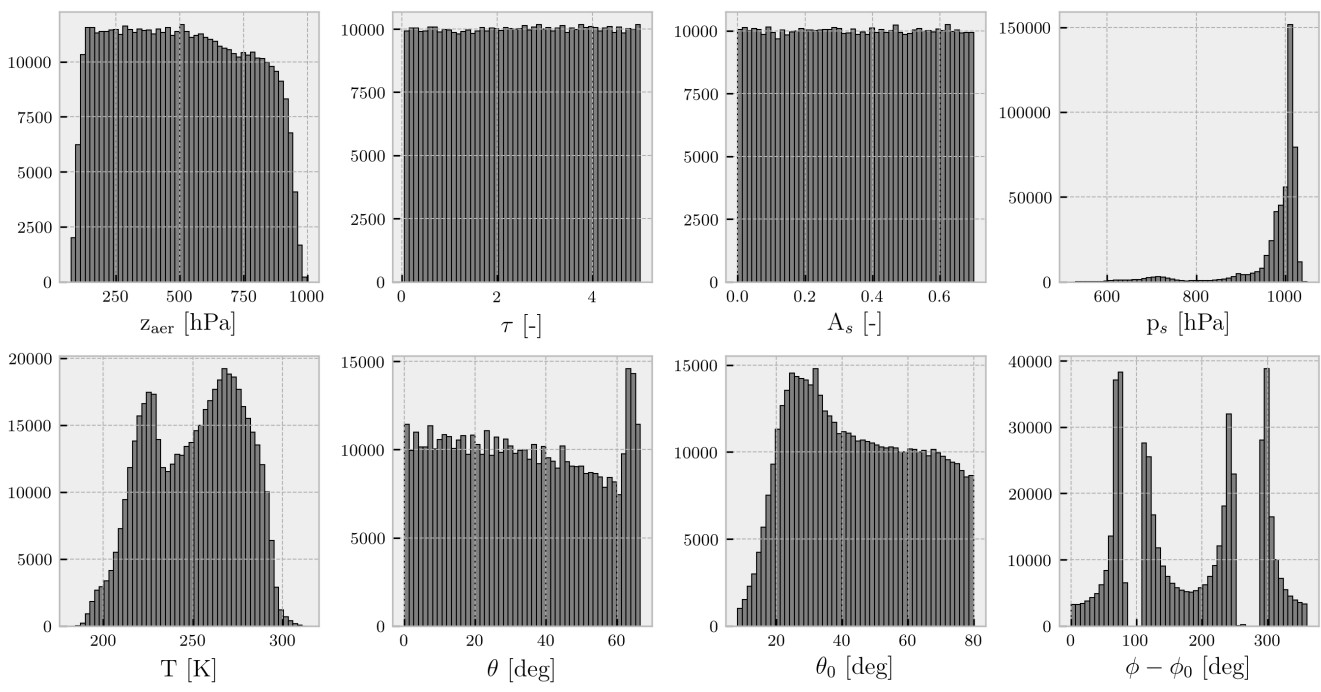

**Figure 1.** Histograms of the various input parameters for each of the neural network models in NN. Minimum and maximum values for each of the parameters are shown in Table 2.

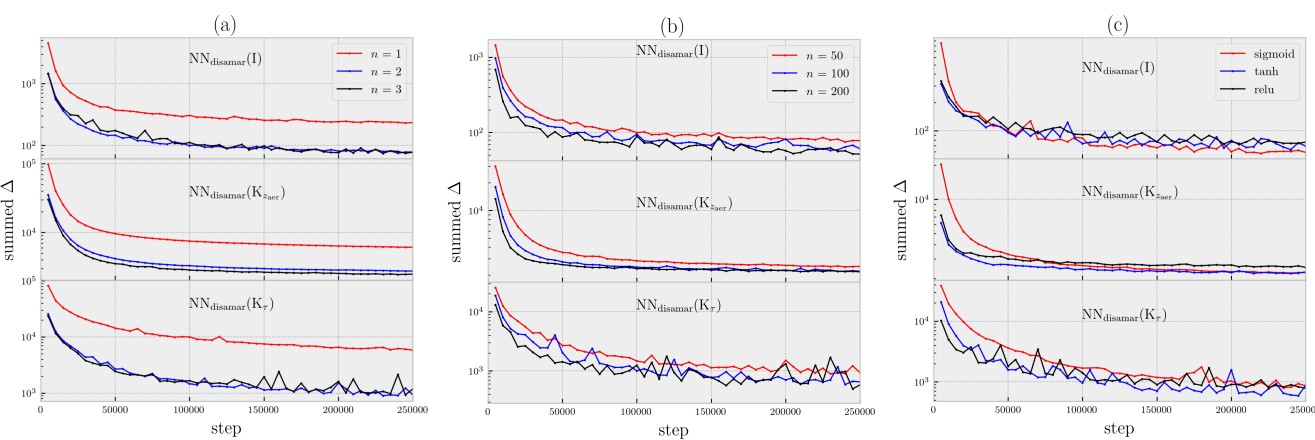

**Figure 2.** Summed loss as a function of training step for different neural network model configurations. **(a)** The neural network models have 50 nodes per each layer with a sigmoid activation function. **(b)** The neural network models have two hidden layers with each node activated by the sigmoid function. **(c)** The neural network models have two hidden layers with a 100 nodes for each layer.

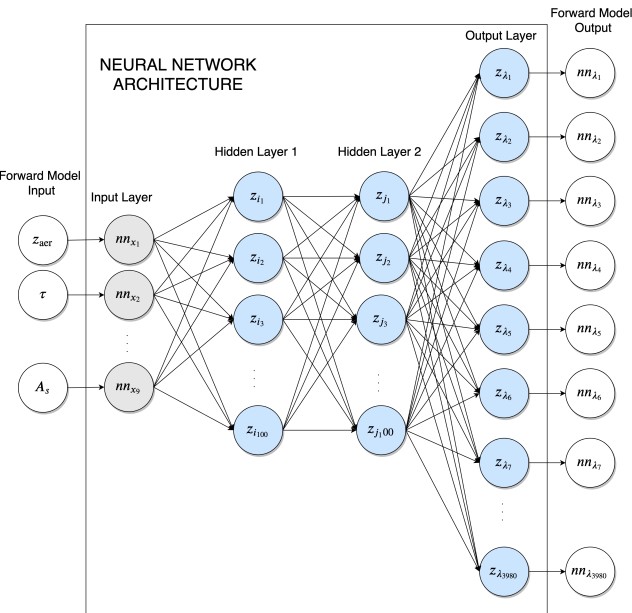

**Figure 3.** Schematic of each of the three neural networks in NN. There are two hidden layers, each containing 100 nodes. $z$ represents inputs for each of the nodes, whereas $nn$ represents the inputs and outputs of the neural network.

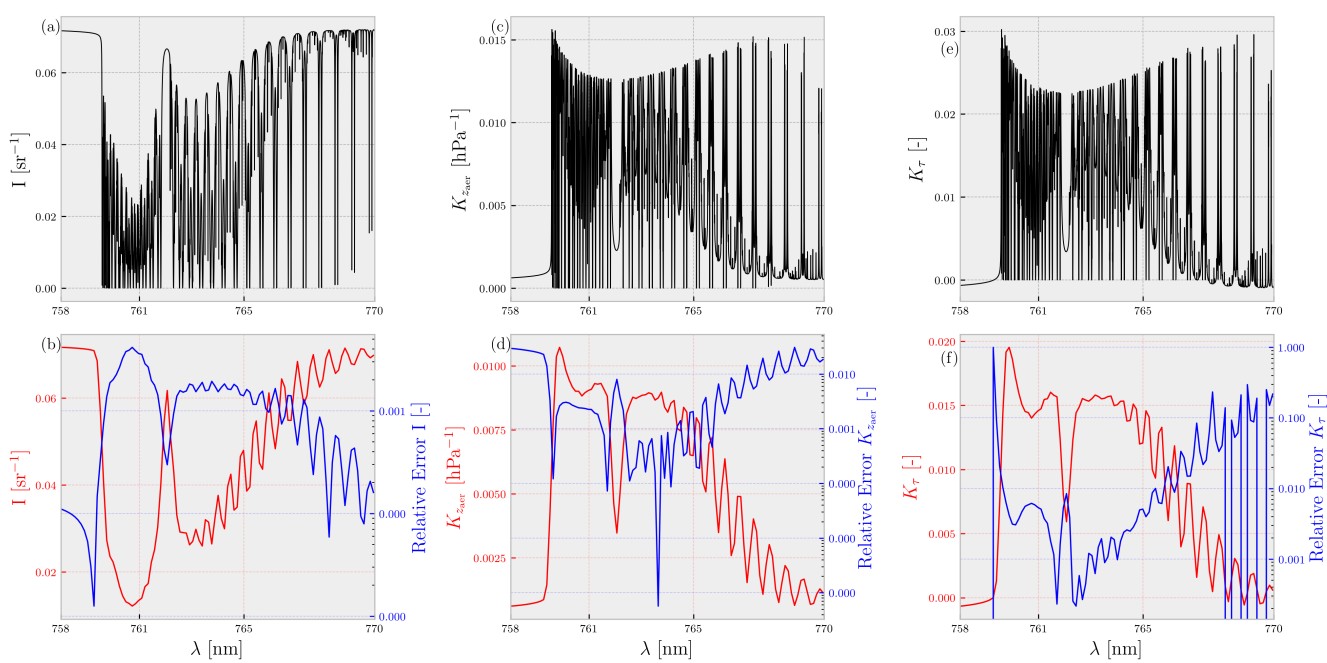

**Figure 4.** Performance of the finalised neural network. The top row represents the averaged output of each of the neural networks for surface albedo less than 0.4. The bottom row represents the convolved version of the top row (plotted as the red line with the left-handed y-axis) and the convolved relative error (plotted in log scale) with the truth (plotted in blue with the right-handed y-axis). The relative errors are computed as the absolute value of the difference (post-convolution) between the averaged true and averaged predicted spectra, divided by the averaged true spectra. (a,b) represent the neural network computed sun-normalised radiances, (c,d) represent the same for the derivative of reflectance with respect to aerosol layer height, and (e,f) the same with respect to aerosol optical thickness.

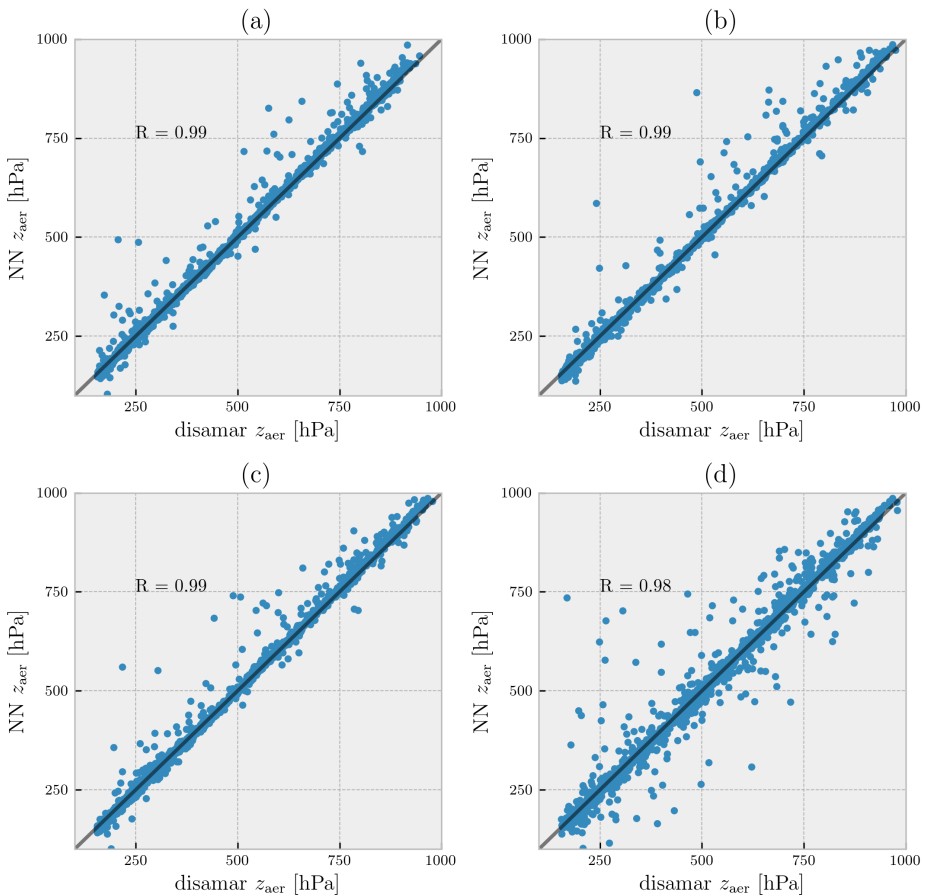

**Figure 5.** Retrieved layer heights compared between Disamar and NN for 2000 synthetic spectra in the presence of model errors. The dots represent converged scenes only, with the x axis representing retrievals from Disamar and the y-axis representing the same from NN. The model errors represented in this figure are (a) aerosol layer pressure thickness, (b) aerosol single scattering albedo, (c) aerosol scattering phase function asymmetry factor, and (d) surface albedo. These results as well as the introduced model errors are summarised in Table 3. The Pearson correlation coefficient (R) between the retrieved $z_{aer}$ from different methods is mentioned in each of the plots.

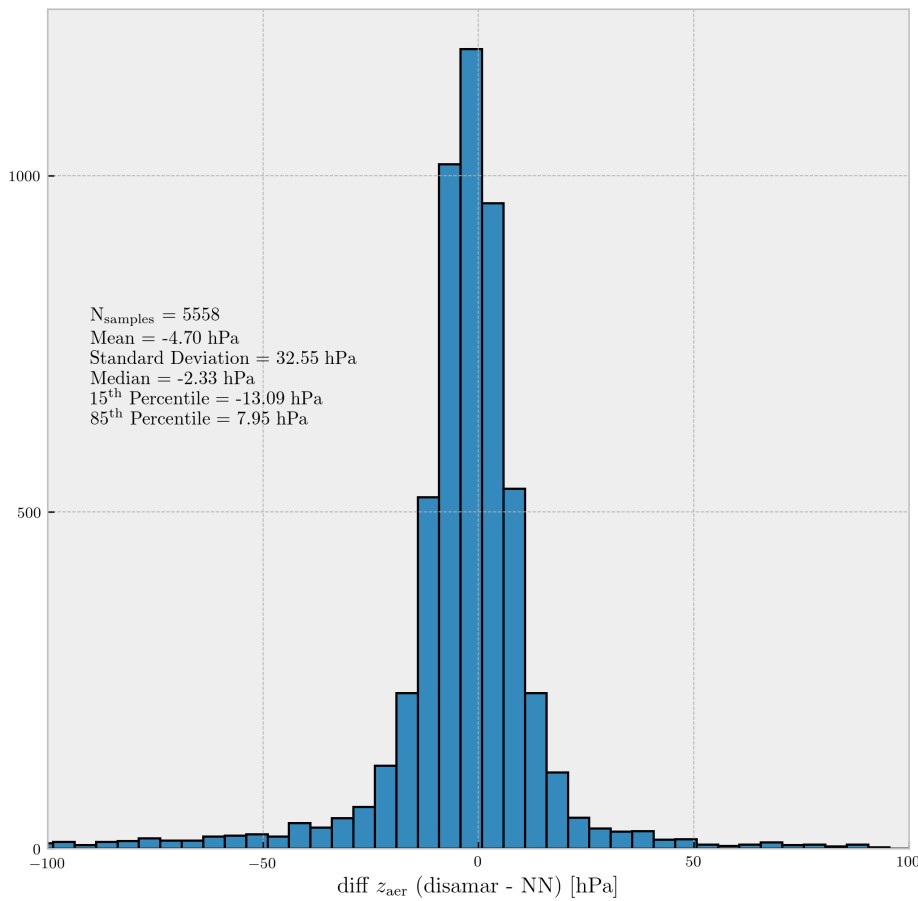

**Figure 6.** Histogram of differences between the retrieved $z_{aer}$ values using Disamar and NN retrieval methods for synthetic spectra generated by Disamar. Total number of cases is 8000, whereas the plot contains 5558 retrieved samples for both Disamar and NN; non-converged cases are not included. A map of these differences are plotted in Figure 9c.

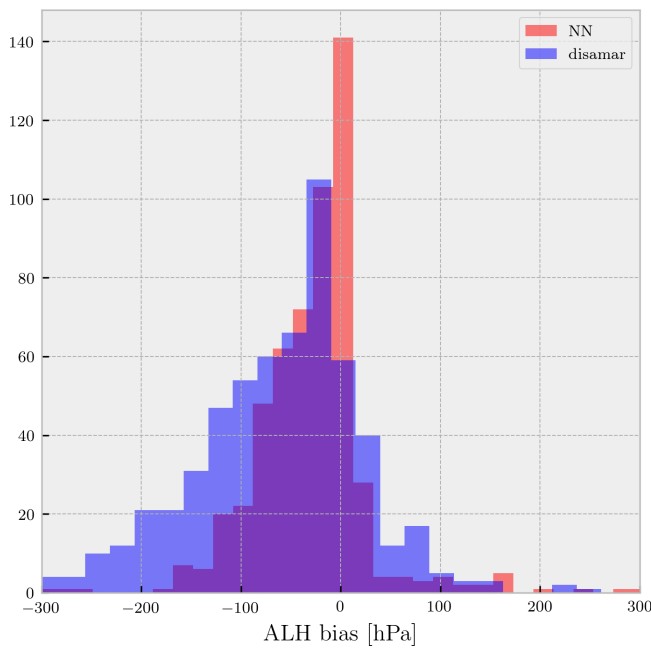

**Figure 7.** Histogram of biases (retrieved - true) for scenes in the synthetic experiment for which either NN converges to a solution (red bar plot) and Disamar does not, or Disamar converges to a solution (blue bar plot) whereas NN does not.

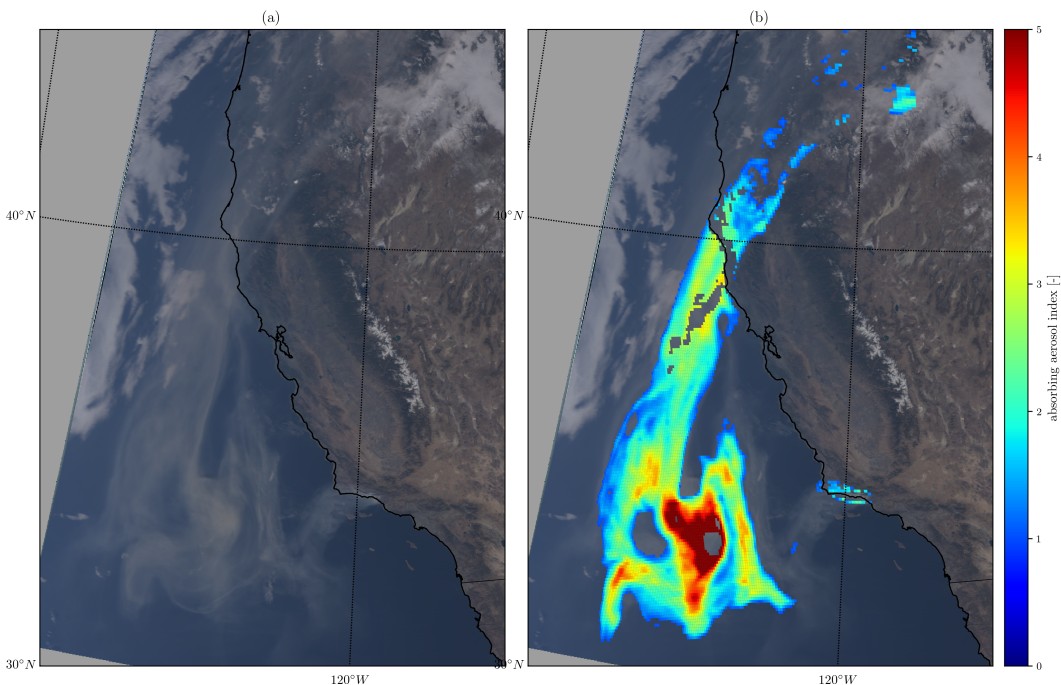

**Figure 8.** (a) MODIS Terra image of the December 12, 2017 Southern Californian wildfire plume, extending from land to ocean. (b) Calculated aerosol absorbing index from the TROPOMI level-2 processor. Missing pixels are flagged by a cloud mask or land-sea mask, or have an AAI less than 1.0.

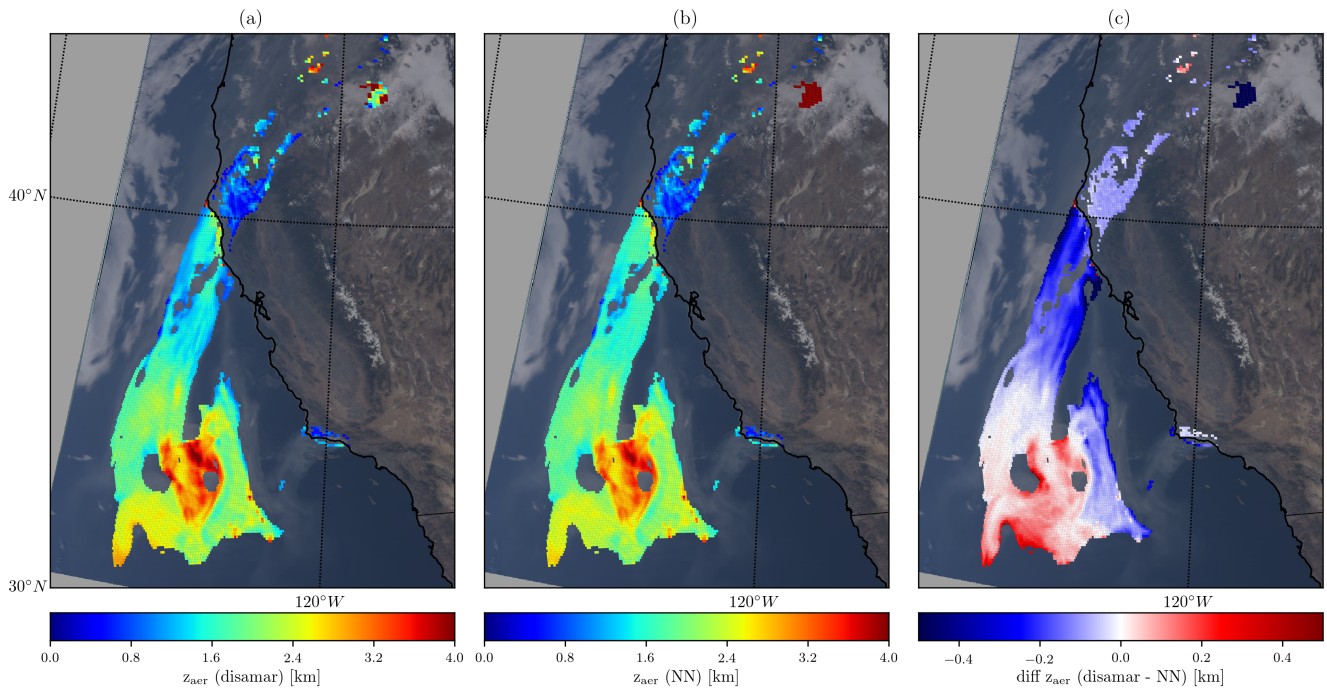

**Figure 9.** (a) Aerosol layer height retrieved using Disamar as the forward model. (b) The same, but with NN replacing Disamar in the operational processor. (c) difference between Disamar and NN retrieved aerosol layer heights.

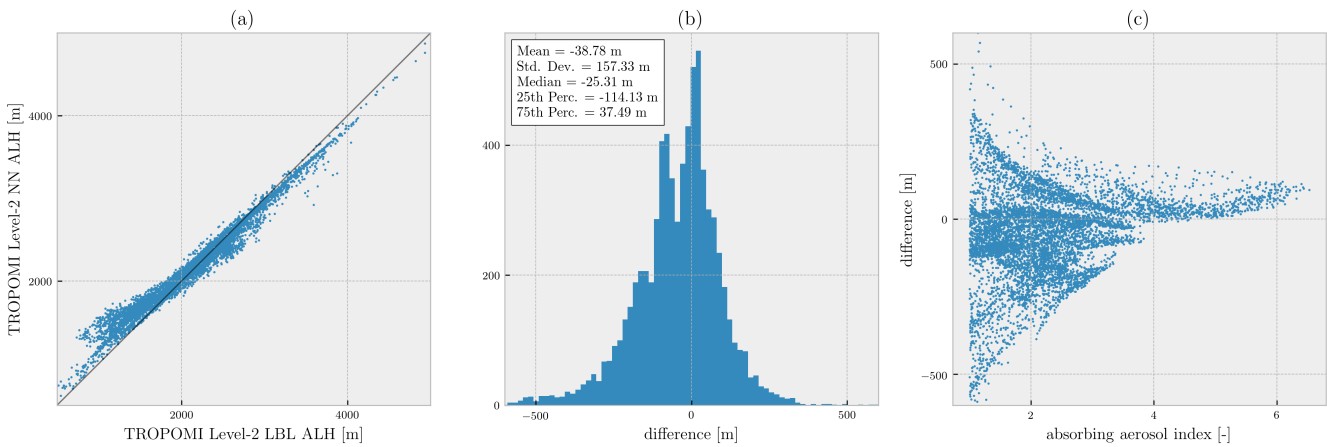

**Figure 10.** Comparison of retrieved aerosol layer heights from TROPOMI-measured spectra (orbit number 858) for the 12th December, 2017 Southern California fires using Disamar and NN. (a) Retrieved aerosol layer heights from the two methods; (b) Histogram of the difference between retrieved heights from Disamar and NN. The difference is defined as $z_{aer}(\text{Disamar})$ - $z_{aer}(\text{NN})$. (c) Differences compared to TROPOMI's operational AAI product (x axis).

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
