# Peer review of "A neural network radiative transfer model approach applied to TROPOMI's aerosol height algorithm"

_Atmospheric Measurement Techniques, 2019_

## Referee Comment (RC1) · Anonymous Referee #1 · 20 Jun 2019

TROPOMI aerosol layer height retrievals use the Disamar radiative transfer model, which performs line-by-line calculations and requires several minutes to retrieve aerosol layer height for a single sounding. This limits the yield of the aerosol layer height processor. The authors propose using artificial neural networks in place of Disamar. They train three neural networks, one for the reflectance and one each for the derivative of the reflectance with respect to aerosol layer height and aerosol optical thickness. The neural network models produced reflectance and Jacobian biases in the 1-3% range. Retrievals were performed for both synthetic and real data. For synthetic data, the neural network retrievals of aerosol height had a median difference of about 2 hPa compared to Disamar retrievals. For a real test case (forest fires in South-

ern California), retrieved aerosol height using the neural network was, on average, less than 50 meters, different from the Disamar value. The neural network retrievals were three orders of magnitude faster than line-by-line retrievals.

General Comments:

This work is novel and interesting. The proposed algorithm produces results that have comparable accuracy to line-by-line models while achieving three orders of magnitude speed-up in computational efficiency. This makes it possible to increase the throughput of TROPOMI aerosol retrievals and possibly enable operational retrievals for all pixels in each TROPOMI orbit. The computational speed-up also opens up the possibility of including more physics in the forward model.

The usage of three neural network models is an interesting idea. It takes advantage of the fact that correlations between input parameters and different forward model outputs are different.

The paper should be published, but only after the comments (see below) are dealt with.

Specific Comments:

Line 14, page 1: "eligible for retrieving aerosol layer height". Is this because of clouds? If this is the case, say so.

Lines 26-28, page 1: The previous sentence suggests that the method utilized line-by-line calculations to generate training data set. Do the authors mean that line-by-line calculations are not used in the "operational" retrieval that utilizes neural networks? The authors also need to say more to distinguish their method from that used by Chimot et al. and Loyola et al.

Line 33, page 1 – Line 1, page 2: It is not clear what the difference is from the existing neural network approaches. Is it that Optimal Estimation is used? "using artificial neural networks to improve the computational speed of RT calculations" is very vague and general; isn't that common to all neural network approaches?

[Figure]

Line 8, page 2: Add the following references:

Timofeyev et al., 1995; Sanghavi et al., 2012; Geddes & Bösch, 2015; Colosimo et al., 2016; Davis et al., 2017; Xu, et al., 2017; Zeng et al., 2018

Colosimo, S. F., V. Natraj, S. P. Sander, and J. Stutz (2016), A sensitivity study on the retrieval of aerosol vertical profiles using the oxygen A-band, Atmos. Meas. Tech., 9(4), 1889–1905, doi:10.5194/amt-9-1889-2016.

Davis, A. B., O. V. Kalashnikova, and D. J. Diner (2017), Aerosol layer height over water from O2 A-band: mono-angle hyperspectral and/or bispectral multi-angle observations, Preprint, doi:10.20944/preprints201710.0055.v1.

Geddes, A., and H. Bösch (2015), Tropospheric aerosol profile information from high-resolution oxygen A-band measurements from space, Atmos. Meas. Tech., 8(2), 859–874, doi:10.5194/amt-8-859-2015.

Sanghavi, S., J. V. Martonchik, J. Landgraf, and U. Platt (2012), Retrieval of aerosol optical depth and vertical distribution using O2 A- and B-band SCIAMACHY observations over Kanpur: A case study, Atmos. Meas. Tech., 5(5), 1099–1119, doi:10.5194/amt-5-1099-2012

Timofeyev, Y. M., A. V. Vasilyev, and V. V. Rozanov (1995), Information content of the spectral measurements of the 0.76 $\mu$m O2 outgoing radiation with respect to the vertical aerosol optical properties, Adv. Space Res., 16(10), 91–94, doi:10.1016/0273-1177(95)00385-R.

Xu, X., Wang, J., Wang, Y., Zeng, J., Torres, O., Yang, Y., et al. (2017). Passive remote sensing of altitude and optical depth of dust plumes using the oxygen A and B bands: First results from EPIC/DSCOVR at Lagrange-1 point, Geophys. Res. Lett., 44, 7544–7554, doi:10.1002/2017GL073939.

Zeng, Z.-C., V. Natraj, F. Xu, T. J. Pongetti, R.-L. Shia, E. A. Kort, et al. (2018), Constraining aerosol vertical profile in the boundary layer using hyperspectral measurements of oxygen absorption, Geophys. Res. Lett., 45, doi:10.1029/2018GL079286.

Line 24-26, page 4: "The polarized . . . 760 nm".

1. First order scattering also has a polarization effect. Presumably, the authors mean that they ONLY compute the first order polarization. If so, please state that.

2. What is "small"? < 5%? < 1%? This statement is potentially untrue. If true, values for how small is small must be given, with proof. In the continuum and in weak lines, the second order effects might be large.

Lines 27-28, page 4: There is a contradiction here. If the exclusion is not advised, the effect cannot be small. The authors should simply state that they ignored this for computational reasons. Besides, the whole point of using neural networks is to speed up calculations. Why not use them for speeding up Raman calculations too, or at least use lookup tables for Raman effects?

Lines 29-30, page 4: "From preliminary . . . significantly".

Quantify this statement, and ideally provide a figure to illustrate the effects.

Lines 30-31, page 4: Although it is true that retrievals are typically performed under "cloud free" conditions, optically thin cirrus clouds need to be accounted for since they are almost always present.

Lines 1-8, page 5: What are the effects of these approximations on the retrieved results? It seems that many of these simplifications are not needed because of the use of neural networks. Also, if only single scattering is used, calculation of ANY phase function is trivial and not time consuming. Considering the fact that the authors aim to produce an operational retrieval algorithm, these simplifications seem unwarranted and restrictive.

Table 2, page 8: What does "varied" mean for the aerosol layer thickness? Is the aerosol layer thickness part of the feature vector? If not, how is it handled?

Line 8, page 8: "a choice of 500,000 Disamar generated spectra"

How are these spectra generated? It is not clear how the choice is made.

Lines 20-22, page 9:

Need more quantitative error information, like for the derivative with respect to tau. Also, what does continuum (3d) mean?

Lines 23-24, page 9: "these parts . . . cross sections"

Why do low oxygen absorption cross sections lead to low aerosol information content?

Technical Comments:

Line 19, page 1: correlative -> correlated

Line 20, page 1: Hasekamp and Butz (2008) -> (Hasekamp and Butz, 2008)

Line 19, page 4: in -> on

Line 25, page 5: an -> a

Line 17, page 7: differentiation which -> differentiation, which

Line 2, page 8: selected TROPOMI -> selected to represent TROPOMI

Line 6, page 8: an -> and

Line 7, page 8: isn't a -> is no

Line 14, page 8: legible -> physical

Line 12, page 9: were trained -> was trained

Line 18, page 9: to -> of

Line 20, page 9: remove "more than"

Line 23, page 9: at -> in

Line 24, page 9: with respect -> compared

Line 16, page 10: less than -> by less than; remove "approximately"

Line 23, page 10: less than -> less by

Line 29, page 10: were -> was

Table 3 caption, page 11: an -> and

Line 13, page 11: agreements, they primarily departed in the -> agreement, they primarily differed for the

Line 16, page 11: departure, different -> bias, differing

Line 17, page 11: departure -> bias

Table 4 caption, page 11: disamar -> Disamar

Figure 1 caption, page 14: available -> shown

Figure 2 caption, page 15: A schematic -> Schematic

Figure 4, page 17: need x axis label for (a), x and y axis labels for (b), correct x and y axis labels for (c), y axis label for (d), correct y axis label for (d)

Figure 5 caption, page 18: A histogram -> Histogram; plotting -> plotted

Figure 6 caption, page 19: A histogram -> Histogram

Figure 7 caption, page 20: A MODIS-> MODIS; remove "the" before "ocean"; remove "either"; cloud mask, or by a land-sea mask -> cloud mask or land-sea mask

Figure 8 caption, page 21: represents the difference -> Difference

Figure 9 caption, page 21: Figre (a) directly compares retrieved aerosol layer heights from the two methods. Figure (b) provides a histogram of the difference between these retrieved heights from Disamar and NN. The difference is defined as zaer(Disamar) -

zaer(NN). Figure (c) compares these differences with TROPOMI's operational absorbing aerosol index product (x axis). -> (a) Retrieved aerosol layer heights from the two methods; (b) Histogram of the difference between retrieved heights from Disamar and NN. The difference is defined as zaer(Disamar) - zaer(NN); (c) Differences compared to TROPOMI's operational absorbing aerosol index product (x axis).

---

## Referee Comment (RC2) · Anonymous Referee #2 · 3 Jul 2019

**General Comments**

Nanda et al. present a method to accelerate radiative transfer calculations based on neural networks (NN), to speed up an optimal-estimation based retrieval of aerosol layer height (ALH) from TROPOMI O2-A band observations. The neural network is trained/validated on a set of simulated TROPOMI spectra. The ALH retrieval using the NN-based forward model is compared to results using an explicit line-by-line forward model. The NN version of the retrieval produces results consistent with the explicit model for a synthetic and real test case, whilst improving the speed by three orders of magnitude.

[Figure]

Overall, I think this paper is interesting and well within the scope of AMT. In Its present form, it is missing some key details (see comments). Once these are addressed I will recommend publishing.

**Specific Comments**

*Page 2 Line 18: "The bottleneck identified..."*

There are a few other commonly used methods for accelerating RT simulations e.g. optical property PCA and low-streams interpolation (see cited review below). It may be worth mentioning why NN is being chosen over these.

Natraj V. (2013) A review of fast radiative transfer techniques. In: Kokhanovsky A. (eds) Light Scattering Reviews 8. Springer Praxis Books. Springer, Berlin, Heidelberg

*Page 4 Line 30: From preliminary tests, the exclusion of RRS....*

Since there were preliminary tests for the sensitivity to RRS, it may be worth mentioning what these were and quantitatively how these impacted $z_{aer}$. Solar-induced chlorophyll fluorescence in the A-Band may also have a similar effect spectrally, and likely has a greater impact on the spectra. See

Frankenberg et al. (2011) Disentangling chlorophyll fluorescence from atmospheric scattering effects in O2 A-Band spectra of reflected sunlight, Geophysical Research Letters, vol 38, L03801 https://doi.org/10.1029/2010GL045896

*Page 4 Line 31: The aerosol fraction is assumed as 1.0*

Could you define what you mean by aerosol fraction?

*Page 5, Line 1-5: Perhaps the largest simplification...*

There are many assumptions here - The aerosol optical properties are fixed (0.95 SSA, g=0.9 using a simplified HG phase function). Either literature justifying why these assumptions are ok must be cited, or the authors need to test these how these assumptions impact the retrieval results e.g. by testing against synthetic data with realistic optical properties. I would be curious to see how the retrievals perform for cases with very different optical properties e.g. dust

*Page 7, Line Line 19: The inputs for the NN are referred together as the feature vector...*

The fixture of the aerosol optical properties in the optimal estimation approach seems quite restrictive. Have you performed an information content analysis of the TROPOMI O2-A band to check if retrieving some of these can reduce the uncertainty in aerosol height? E.g. allowing the SSA to vary may reduce the potential influence on its parameter error inducing a corresponding ALH error.

*Page 7, Line 28: "whereas NN only uses the temperature at $z_{aer}$"*

*For the meteorological parameters, is there any rationale for excluding other potentially important predictors from being included e.g. PBL height or surface heating fluxes and wind speeds that may also provide prior information about the ALH.*

*Page 8 Section 3.2 First Paragraph*

*From my reading of this, the profiles are generated randomly after selecting a random set of tropomi solar-viewing geometry combinations. Naively, I would expect that the most representative way to create the training set would be to select the profiles from the ERA reanalysis corresponding to the randomly selected orbit geometries - the model probably doesn't need to reproduce a spectrum of Saharan dust for a typical Antarctic viewing geometry. Perhaps there is a heuristic argument that the way you are doing it could more reliably span the entire set of profile/viewing combinations, but if this is so you should state it in the manuscript. Ideally you would want to test the performance by comparing multiple approaches of generating the training data, though I am not sure of your computational resources*

*Page 9, Line 1: Finding the most optimal neural network configuration requires...*

*Using a single set for validation leads to questions about robustness in the validation.*

*Typically cross validation e.g. k-fold methods are used to derive more accurate esti-mates of model performance. Have you looked into how robust the single training set is?*

*Page 9, Line 6: The sigmoid function is chosen for activation...*

*Can you show evidence for the sigmoid function outperforming the other forms?*

*Page 9, Line 8: For each of the neural network models. . .*

*It would be useful showing a comparison of performance for the different models tested - I don't have an intuitive sense from the description of how different the actual results were for the different layer/node combinations, or how robust one instance of a 25000 iteration training is. For instance if I retrained the two layer model with 100+100 nodes again with a different initialization, would it still be the optimum configuration?*

*Page 11, Line 12: Although the retrieval algorithms have good agreements. . .*

*For the low aerosol loading scenes, what happens when you include both surface albedo values in the NN model?*

***Minor Corrections***

*Equation 2: Bold the x in the forward model to be consistent with notation for vectors.*

*Page 7, Line 17: Change "automatic differentiation which is a powerful algorithm that computes" to "automatic differentiation which computes"*
* * *
*Interactive comment on Atmos. Meas. Tech. Discuss., doi:10.5194/amt-2019-143, 2019.*

---

## Author Comment (AC1) · 11 Jul 2019

**Reviewer comment (general):** This work is novel and interesting. The proposed algorithm produces results that have comparable accuracy to line-by-line models while achieving three orders of magnitude speed-up in computational efficiency. This makes it possible to increase the throughput of TROPOMI aerosol retrievals and possibly enable operational retrievals for all pixels in each TROPOMI orbit. The computational speed-up also opens up the possibility of including more physics in the forward model. The usage of three neural network models is an interesting idea. It takes advantage of the fact that correlations between input parameters and different forward model outputs

are different. The paper should be published, but only after the comments (see below) are dealt with.

**Author's response:** Thank you for taking the time to review this manuscript. The goal of this research project was to develop a faster aerosol layer height retrieval algorithm, while keeping in mind the possibility of including more details into the model in the future.

**Reviewer comment (specific 1):** Line 14, page 1: "eligible for retrieving aerosol layer height". Is this because of clouds? If this is the case, say so.

**Author's response:** The rough estimate of 3% of total eligible pixels for retrieving aerosol layer height needs to be updated following new analyses of 56 TROPOMI files over Europe containing 7.3 million pixels. The selection criterion was a UV Aerosol Indices above 0.0, of which 6.1% of all pixels considered over Europe. TROPOMI's UV Aerosol Index values are one index point lower than UV Aerosol Indices from other indices.

**Changes to the manuscript:** The new sentence now reads the following:

'With TROPOMI recording approximately 1.4 million pixels within a single orbit, a rough estimate based on a minimum UV Aerosol Index of 0 indicates that at least six percent of all pixels over an area as large as Europe will be eligible for retrieving aerosol layer height. This number can go beyond 50,000 pixels per orbit in many cases, placing a steep requirement on the computational infrastructure to process all possible pixels from a single orbit.'

**Reviewer comment (specific 2):** Lines 26-28, page 1: The previous sentence suggests that the method utilized line-by-line calculations to generate training data set. Do

the authors mean that line-by-line calculations are not used in the "operational" retrieval that utilizes neural networks? The authors also need to say more to distinguish their method from that used by Chimot et al. and Loyola et al.

**Reviewer comment (specific 3):** Line 33, page 1 – Line 1, page 2: It is not clear what the difference is from the existing neural network approaches. Is it that Optimal Estimation is used? "using artificial neural networks to improve the computational speed of RT calculations" is very vague and general; isn't that common to all neural network approaches?

**Author's response:** The work by Chimot et al. utilise the DISAMAR radiative transfer model to compute synthetic OMI measurements of the slant column density in the $O_2$-$O2$ absorption band, which is a part of the feature vector in the neural network models in their implementation of retrieving aerosol layer height. Chimot et al. do not use any line-by-line calculations in their operational retrievals.

The aerosol layer height algorithm that is the subject of this paper follows a similar philosophies to Chimot et al. as well as Loyola et al, with important differences.

- With respect to Chimot et al. the paper discusses using DISAMAR to generate synthetic spectra for training a neural network model, the difference being that while Chimot et al. prefer to retrieve aerosol layer height as the output of their trained neural network model, whereas the neural network model in the paper outputs the top-of-atmosphere oxygen A-band spectra in the forward model. These neural-network-model-calculated top-of-atmosphere oxygen A-band spectra are then utilised by an optimal estimation scheme, which outputs a retrieved aerosol layer height value.

- With respect to Loyola et al, the neural network models both compute top-of-atmosphere spectra, with the difference being that the KNMI aerosol layer height neural network retrieval algorithm has two other neural network models for the

derivatives of the spectra with respect to the state vector parameters aerosol optical thickness and aerosol layer height, whereas Loyola et al. train their neural network models only for the sun-normalised radiances (section 4.4, https://www. atmos-meas-tech.net/11/409/2018/amt-11-409-2018.pdf)

**Changes to the manuscript:**

- Adjusted text to 'Chimot et al. (2017) describe an approach using a radiative transfer model to generate OMI slant column densities of the $O_2$-$O2$ band at 477 nm for different aerosol optical depths (among other input parameters) to train several artificial neural network models that directly retrieve aerosol layer height. Operationally, their neural network models use the MODIS aerosol optical depth at 550 nm product and retrieved OMI slant column densities, thereby entirely foregoing line-by-line calculations and significantly speeding up the retrieval algorithm.'.

- Amended the final paragraph of section 1 to 'The work of Chimot et al. (2017) and Loyola et al. (2018) bring to light the efficacy of artificial neural networks in satellite remote sensing of oxygen absorption bands for retrieving properties of scattering species in the atmosphere. This paper discusses a method inspired by Chimot et al. and Loyola et al. to retrieve aerosol layer height from oxygen A-band measurements by TROPOMI. While Chimot et al. directly retrieve aerosol layer heights from their neural network models, the operational algorithm in this paper utilises neural networks to calculate top-of-atmosphere radiances in the forward model. This is subsequently used by an optimal estimation scheme to retrieve aerosol layer heights. Similarly while Loyola et al. derive top-of-atmosphere sun-normalised radiances only for their cloud property retrieval algorithm, the method in this paper has dedicated neural network models that calculate the Jacobian as well as the top-of-atmosphere sun-normalised radiances.'.

[Figure]

**Reviewer comment (specific 4):** Line 8, page 2: Add the following references:

Timofeyev et al., 1995; Sanghavi et al., 2012; Geddes Bösch, 2015; Colosimo et al., 2016; Davis et al., 2017; Xu, et al., 2017; Zeng et al., 2018

Colosimo, S. F., V. Natraj, S. P. Sander, and J. Stutz (2016), A sensitivity study on the retrieval of aerosol vertical profiles using the oxygen A-band, Atmos. Meas. Tech., 9(4), 1889–1905, doi:10.5194/amt-9-1889-2016.

Davis, A. B., O. V. Kalashnikova, and D. J. Diner (2017), Aerosol layer height over water from O2 A-band: mono-angle hyperspectral and/or bispectral multi-angle observations, Preprint, doi:10.20944/preprints201710.0055.v1.

Geddes, A., and H. Bösch (2015), Tropospheric aerosol profile information from high resolution oxygen A-band measurements from space, Atmos. Meas. Tech., 8(2), 859–874, doi:10.5194/amt-8-859-2015.

Sanghavi, S., J. V. Martonchik, J. Landgraf, and U. Platt (2012), Retrieval of aerosol optical depth and vertical distribution using O2 A- and B-band SCIAMACHY observations over Kanpur: A case study, Atmos. Meas. Tech., 5(5), 1099–1119, doi:10.5194/amt-5-1099-2012

Timofeyev, Y. M., A. V. Vasilyev, and V. V. Rozanov (1995), Information content of the spectral measurements of the 0.76 $\mu$m O2 outgoing radiation with respect to the vertical aerosol optical properties, Adv. Space Res., 16(10), 91–94, doi:10.1016/0273-1177(95)00385-R.

Xu, X., Wang, J., Wang, Y., Zeng, J., Torres, O., Yang, Y., et al. (2017). Passive remote sensing of altitude and optical depth of dust plumes using the oxygen A and B bands: First results from EPIC/DSCOVR at Lagrange-1 point, Geophys. Res. Lett., 44, 7544–7554, doi:10.1002/2017GL073939.

Zeng, Z.-C., V. Natraj, F. Xu, T. J. Pongetti, R.-L. Shia, E. A. Kort, et al. (2018), Constraining aerosol vertical profile in the boundary layer using hyperspectral measurements of oxygen absorption, Geophys. Res. Lett., 45, doi:10.1029/2018GL079286.

**Author's response:** Agreed.

**Changes to the manuscript:** Amended the first sentence of section 2 that discusses previous work done for retrieving vertical information of aerosols using passive space-borne measurements of the oxygen A-band to 'The TROPOMI aerosol layer height is one of the many algorithms that exploit vertical information of scattering aerosol species in the oxygen A-band (Timofeyev et al., 1995; Gabella et al., 1999; Corradini and Cervino, 2006; Pelletier et al., 2008; Dubuisson et al., 2009; Frankenberg et al., 2012; Wang et al., 2012, Sanghavi et al., 2012; Sanders and de Haan, 2013; Hollstein and Fischer, 2014; Geddes Bösch, 2015; Sanders et al., 2015; Colosimo et al., 2016; Sanders and de Haan, 2016; Davis et al., 2017; Xu, et al., 2017; Zeng et al., 2018; Nanda et al., 2018b)'

**Reviewer comment (specific 5):** Line 24-26, page 4: "The polarized . . . 760 nm".

- First order scattering also has a polarization effect. Presumably, the authors mean that they ONLY compute the first order polarization. If so, please state that.

- What is "small"? $< 5\%$? $< 1\%$? This statement is potentially untrue. If true, values for how small is small must be given, with proof. In the continuum and in weak lines, the second order effects might be large.

**Reviewer comment (specific 6):** Lines 27-28, page 4: There is a contradiction here. If the exclusion is not advised, the effect cannot be small. The authors should simply state that they ignored this for computational reasons. Besides, the whole point of using neural networks is to speed up calculations. Why not use them for speeding up Raman calculations too, or at least use lookup tables for Raman effects?

**Reviewer comment (specific 7):** Lines 29-30, page 4: "From preliminary . . . significantly".

Quantify this statement, and ideally provide a figure to illustrate the effects.

**Author's response:**

- Polarisation is ignored in the sense that for retrieving aerosol layer heights DISAMAR only computes the first element of the Stoke's vector in the radiation fields. The exclusion of higher order Stoke's vector elements has not shown to be a significant source of error.

- The affect of ignoring Rotational Raman Scattering (RRS) in the forward model results in errors in the final retrieved aerosol layer heights. However, as clarified by Sanders and de Haan (2016) who have retrieved aerosol layer height using the same radiative transfer model while including and excluding RRS, this error is significantly less in comparison to other model errors. Because the inclusion of RRS has resulted in a significant increase in time required by the line-by-line radiative transfer model and ignoring it does not yield large errors (from synthetic experiments), it has been historically excluded in the KNMI aerosol layer height retrieval algorithm.

With regards to the reviewer's suggestions to use lookup tables for the Raman effects or potentially incorporating an artificial neural network solution for including RRS into the forward model calculations, the authors appreciate these ideas very much. However, since the goal of this paper is to create a model that replicates the existing TROPOMI aerosol layer height algorithm, RRS is ignored for the sake of comparison and benchmarking. In the future, this may be a serious consideration by the TROPOMI Level-2 algorithm development team.

Finally, the authors acknowledge the confusion in this paragraph. The sentence 'RRS

can alter the line depths in the O2 A-band, but this effect is small', is not complete. The authors meant to state that the effect of excluding RRS on the retrieved aerosol layer height is small. The changes to the manuscript will reflect this. With regards to the reviewer's comments on quantifying the statement, the authors have chosen to include a citation to Sanders and de Haan 2016, which is the Algorithm Theoretical Basis Document of the TROPOMI ALH retreival algorithm. This discusses the rationale behind excluding RRS from computations.

**Changes to the manuscript:** To address the questions raised by the reviewer, the following amendments have been done to Section 2.2, paragraph 3 of the manuscript.

- 'As the Rayleigh optical thickness is low at 760 nm, DISAMAR only computes the monochromatic component of light by calculating the first element of the Stoke's vector. The exclusion of higher order Stoke's vector elements of the radiation fields has not shown to be a significant source of error (Sanders and de Haan, 2016).'

- 'While this exclusion of RRS is not advised by literature (Sioris and Evans, 2000; Vasilkov et al. 2013), preliminary experiments by Sanders and de Haan (2016) have ascertained that the errors in the retrieved aerosol layer height resulting from ignoring RRS of the oxygen A-band in the forward model are significantly smaller than the effect of other model errors. Due to this, the KNMI aerosol layer height retrieval algorithm has historically ignored calculating RRS cross sections.'

**Reviewer comment (specific 8):** Lines 30-31, page 4: Although it is true that retrievals are typically performed under "cloud free" conditions, optically thin cirrus clouds need to be accounted for since they are almost always present.

**Author's response:** It is indeed correct that optically thin cirrus clouds need to be

accounted for as they are almost always present in the scene. Currently however, there are no implementations in the algorithm to incorporate cirrus cloud properties into the radiative transfer calculations. The operational TROPOMI algorithm utilises a VIIRS cloud mask to flag potential pixels with clouds.

**Changes to the manuscript:** Added the following sentence: 'While optically thin cirrus layers are a known source of error in the retrieved aerosol layer height, currently there are no implementations to tackle this problem. Instead, TROPOMI incorporates information from the VIIRS instrument to detect the presence of clouds in the measured scene.'

**Reviewer comment (specific 9):** Lines 1-8, page 5: What are the effects of these approximations on the retrieved results? It seems that many of these simplifications are not needed because of the use of neural networks. Also, if only single scattering is used, calculation of ANY phase function is trivial and not time consuming. Considering the fact that the authors aim to produce an operational retrieval algorithm, these simplifications seem unwarranted and restrictive.

**Author's response:** It is indeed correct that the simplifications are unwarranted and restrictive for a retrieval algorithm that incorporates a fast neural network approach to replace a radiative transfer model. However the goal of the paper is to replicate (as much as possible) the operational algorithm that uses online line-by-line calculations, which incorporates these approximations to reduce computational time. Finally, the paper compares the retrieved aerosol layer heights from both operational algorithm implementations in order to establish an acceptable agreement between the neural network approach to the online line-by-line approach. This is the first benchmark of the neural-network-augmented retrieval algorithm, subsequently leading to further improvements in the future in line with the reviewer's recommendations.

The affects of these approximations are discussed in detail by Sanders and de Haan

(2016), which is the Algorithm Theoretical Basis Document of the TROPOMI aerosol layer height algorithm. The amendment in the manuscript will reflect their work.

**Changes to the manuscript:** Added the following final paragraph after the mentioned simplifications.

'These simplifications in the DISAMAR forward model are a necessity for the line-by-line aerosol layer height algorithm, owing to its slow computational speed. In contrast, a neural network model is significantly faster. While the speed of the neural network model encourages increasing the complexity of the model, for a comparative study the neural network models are trained to replicate, as best as possible, the line-by-line version. Once this is achieved, the improvement of the algorithm will be an iterative endeavour.'

**Reviewer comment (specific 10):** Table 2, page 8: What does "varied" mean for the aerosol layer thickness? Is the aerosol layer thickness part of the feature vector? If not, how is it handled?

**Author's response:** Aerosol layer thickness is not a part of the feature vector. It is a part of the training data set, and the aerosol layer pressure thickness varies between 50 hPa and 200 hPa. Currently, there is no call by the neural network model to the aerosol layer thickness. This shall be implemented into a future release.

**Changes to the manuscript:** Amended the table entries in Table 2 for aerosol layer thickness. The remark column now reads 'varied but excluded from feature vector', whereas the limits now read '50 hPa - 200 hPa'.

**Reviewer comment (specific 11):** Line 8, page 8: "a choice of 500,000 Disamar generated spectra"

How are these spectra generated? It is not clear how the choice is made.

**Author's response:** The amendment will reflect the clarification of spectra generation.

**Changes to the manuscript:** Changed the sentence 'Following testing and scrutinizing forward model calculation accuracy, a choice of 500,000 Disamar generated spectra is finalised as the size of the training data set.' to the following.

'The number of spectra generated for the training set was determined by training different models with different number of spectra in the training set ranging from 1,000 to 600,000. In general it was observed that incorporating more data resulted in a better neural network model. In order to test the trained neural network model, a choice of 500,000 spectra were selected, and 100,000 spectra were set aside for the test set. These spectra were generated using Disamar with model parameter ranges described in Table 2 and Figure 1.'

To that extent, the following line is removed from Page 6, line 3-5, as there is an incorrect reference to the correct table.

'Finding the most optimal neural network configuration requires a test data set which in this case contains 100,000 scenes outside the training data set. These test data follow the same input model 5 parameter distributions as described in Figure 1 and Table 1.'

**Reviewer comment (specific 12):** Lines 20-22, page 9: Need more quantitative error information, like for the derivative with respect to tau. Also, what does continuum (3d) mean?

**Reviewer comment (specific 13):** Lines 23-24, page 9: "these parts . . . cross sections"

Why do low oxygen absorption cross sections lead to low aerosol information content?

**Author's response:** The following amendment to the text clarifies the role of oxygen

absorption cross sections and aerosol information content.

**Changes to the manuscript:** The following change has been added to the text in the final lines of the final paragraph of section 3.2.

'The neural network model for the derivative of the reflectance with respect to $\tau$ and $z_{\text{aer}}$ perform well in general for parts of the spectrum with large oxygen absorption cross sections, where the value of the derivatives are high (indicating a higher amount of information content from those specific wavelength regions). Errors in the deepest part of the R-branch between 759 nm and 762 nm and the P-branch between 752.50 nm and 765 nm, do not exceed more than 3% for $\text{NN}_{K_{z_{\text{aer}}}}$. The same can be said for $\text{NN}_{K_{\tau}}$, which displays errors in the range of 1% in the same wavelength region. For wavelengths outside of the deepest parts of the R and P-branch, the relative errors are large, and exceed 10% easily. However, the relative errors are calculated as the absolute value of the difference between the true spectrum and the neural network calculated spectrum, divided by the true spectrum. These values can be very large when the value of the true spectrum is very small, which is the case for the derivatives outside the deepest part of the R and P branches. The consequence of these errors in a retrieval scenario from synthetic and real spectra are discussed in the following section.'

**Reviewer comment (technical):**

Line 19, page 1: correlative → correlated

Line 20, page 1: Hasekamp and Butz (2008) → (Hasekamp and Butz, 2008)

Line 19, page 4: in → on

Line 25, page 5: an → a

Line 17, page 7: differentiation which → differentiation, which
Line 2, page 8: selected TROPOMI → selected to represent TROPOMI

Line 6, page 8: an → and

Line 7, page 8: isn't a → is no

Line 14, page 8: legible → physical

Line 12, page 9: were trained → was trained

Line 18, page 9: to → of

Line 20, page 9: remove "more than"

Line 23, page 9: at → in

Line 24, page 9: with respect → compared

Line 16, page 10: less than → by less than; remove "approximately"

Line 23, page 10: less than → less by

Line 29, page 10: were → was

Table 3 caption, page 11: an → and

Line 13, page 11: agreements, they primarily departed in the → agreement, they primarily differed for the

Line 16, page 11: departure, different → bias, differing

Line 17, page 11: departure → bias

Table 4 caption, page 11: disamar → Disamar

Figure 1 caption, page 14: available → shown

Figure 2 caption, page 15: A schematic → Schematic

Figure 4, page 17: need x axis label for (a), x and y axis labels for (b), correct x and y

axis labels for (c), y axis label for (d), correct y axis label for (d)

Figure 5 caption, page 18: A histogram → Histogram; plotting → plotted

Figure 6 caption, page 19: A histogram → Histogram

Figure 7 caption, page 20: A MODIS → MODIS; remove "the" before "ocean"; remove "either"; cloud mask, or by a land-sea mask → cloud mask or land-sea mask

Figure 8 caption, page 21: represents the difference → Difference

Figure 9 caption, page 21: Figre (a) directly compares retrieved aerosol layer heights from the two methods. Figure (b) provides a histogram of the difference between these retrieved heights from Disamar and NN. The difference is defined as zaer(Disamar) - zaer(NN). Figure (c) compares these differences with TROPOMI's operational absorbing aerosol index product (x axis). → (a) Retrieved aerosol layer heights from the two methods; (b) Histogram of the difference between retrieved heights from Disamar and NN. The difference is defined as zaer(Disamar) - zaer(NN); (c) Differences compared to TROPOMI's operational absorbing aerosol index product (x axis).

**Author's response:** Agreed.

**Changes to the manuscript:** Amended the document as requested by the reviewer.

---

## Author Comment (AC2) · 11 Jul 2019

**Reviewer comment (general):**

Nanda et al. present a method to accelerate radiative transfer calculations based on neural networks (NN), to speed up an optimal-estimation based retrieval of aerosol layer height (ALH) from TROPOMI O2-A band observations. The neural network is trained/validated on a set of simulated TROPOMI spectra. The ALH retrieval using the NN-based forward model is compared to results using an explicit line-by-line forward model. The NN version of the retrieval produces results consistent with the explicit model for a synthetic and real test case, whilst improving the speed by three orders of

magnitude.

Overall, I think this paper is interesting and well within the scope of AMT. In Its present form, it is missing some key details (see comments). Once these are addressed I will recommend publishing.

**Author's response:** Thank you for the constructive criticism and for taking the time to review this manuscript. The response to reviewer comments are addressed in the following.

**Reviewer comment (specific 1):** Page 2 Line 18: "The bottleneck identified. . ."

There are a few other commonly used methods for accelerating RT simulations e.g. optical property PCA and low-streams interpolation (see cited review below). It may be worth mentioning why NN is being chosen over these.

Natraj V. (2013) A review of fast radiative transfer techniques. In: Kokhanovsky A. (eds) Light Scattering Reviews 8. Springer Praxis Books. Springer, Berlin, Heidelberg

**Author's response:** We have not tried methods such as PCA or low-streams interpolation to check whether or not artificial neural networks are better at approximating the full physics radiative transfer model.

This is a valid suggestion and related to the remarks by Reviewer 1 on different methods (**amt-2019-143-RC1** Reviewer specific comments 2 and 3) where, in our response and our manuscript amendments, we explain the differences between their approaches and ours. Our main goal was to use a method that can very quickly estimate the reflectance as well as the Jacobian, which is used in the optimal estimation retrieval method. NN turned out to be very efficient for this. It is not known whether other methods, PCA or low-stream interpolations, are capable of doing this. As far as we know, this has never been attempted. We have also looked into k-distribution method, but this turned out to be extremely complex when the Jacobians also had to be included

and therefore abandoned.

**Reviewer comment (specific 2):** Page 4 Line 30: From preliminary tests, the exclusion of RRS. . ..

Since there were preliminary tests for the sensitivity to RRS, it may be worth mentioning what these were and quantitatively how these impacted zaer. Solar-induced chlorophyll fluorescence in the A-Band may also have a similar effect spectrally, and likely has a greater impact on the spectra. See

Frankenberg et al. (2011) Disentangling chlorophyll fluorescence from atmospheric scattering effects in O2 A-Band spectra of reflected sunlight, Geophysical Research Letters, vol 38, L03801 https://doi.org/10.1029/2010GL045896

**Author's response:** The issue on quantifying the sensitivity to RRS has been addressed in the Author's response to Reviewer comment (specific 7) in **amt-2019-143-RC1**.

With regards to the influence of solar-induced chlorophyll fluorescence in the A-band, this is a feature that will be considered in the future release of the TROPOMI aerosol layer height algorithm, as the current focus is to incorporate a neural network model as a replacement for DISAMAR (which is the main line-by-line model). Due to this, the neural network models mimic the operational DISAMAR model (which currently excludes fluorescence) used for retrieving aerosol layer height from TROPOMI measurements as much as possible.

**Changes to the manuscript:**

- With regards to the reviewer's comment on including preliminary tests for the sensitivity of the retrieval to the exclusion of RRS, Section 2.2, paragraph 3 of the manuscript is amended as follows.

[Figure]

'While this exclusion of RRS is not advised by literature (Sioris and Evans, 2000; Vasilkov et al. 2013), preliminary experiments by Sanders and de Haan (2016) have ascertained that the errors in the retrieved aerosol layer height resulting from ignoring RRS of the oxygen A-band in the forward model are significantly smaller than the effect of other model errors. Due to this, the KNMI aerosol layer height retrieval algorithm has historically ignored calculating RRS cross sections.'

- With regards to the reviewer's comment on the impact of chlorophyll, the following amendment has been added to the final paragraph of section 2.2.

'The surface is assumed to be an isotropic reflector with a brightness described by its Lambertian Equivalent Reflectivity (LER). This is also an important simplification, requiring less computations over other surface models such as a Bi-directional Reflectance Model. Although the forward model is capable of including sun-induced chlorophyll fluorescence into the retrieval, it is currently being considered for a future implementation of TROPOMI's operational ALH retrieval algorithm. Lastly, the atmosphere is spherically corrected for incoming solar radiation and remains plane-parallel for outgoing Earth radiance.'

**Reviewer comment (specific 3):** Page 4 Line 31: The aerosol fraction is assumed as 1.0

Could you define what you mean by aerosol fraction?

**Author's response:** Agreed.

**Changes to the manuscript:** With respect to the reviewer's comment, the line in contention has been amended to:

'The fraction of the pixel containing aerosols is assumed to be 100%, which further simplifies the representation of aerosols within the atmosphere.'

**Reviewer comment (specific 4):** Page 5, Line 1-5: Perhaps the largest simplification...

There are many assumptions here - The aerosol optical properties are fixed (0.95 SSA, g=0.9 using a simplified HG phase function). Either literature justifying why these assumptions are ok must be cited, or the authors need to test these how these assumptions impact the retrieval results e.g. by testing against synthetic data with realistic optical properties. I would be curious to see how the retrievals perform for cases with very different optical properties e.g. dust

**Author's response:** These fixed aerosol optical properties have been derived from AERONET data, and the consequences of fixing them are discussed by Sanders et al. (2015), who use GOME-2 and SCIAMACHY spectra to show that these model assumptions aren't the main source of error.

**Changes to the manuscript:** The following sentence and citation has been added to the end of line 5 in page 5.

'These fixed aerosol optical properties have been derived from AERONET data and the consequences of fixing them are discussed by Sanders et al. (2015), who used GOME-2 spectra to show that the algorithm is robust against these model assumptions.'

**Reviewer comment (specific 5):** Page 7, Line Line 19: The inputs for the NN are referred together as the feature vector...

The fixture of the aerosol optical properties in the optimal estimation approach seems quite restrictive. Have you performed an information content analysis of the TROPOMI O2-A band to check if retrieving some of these can reduce the uncertainty in aerosol height? E.g. allowing the SSA to vary may reduce the potential influence on its parameter error inducing a corresponding ALH error.

**Reviewer comment (specific 6):** Page 7, Line 28: "whereas NN only uses the temperature at zaer"

For the meteorological parameters, is there any rationale for excluding other potentially important predictors from being included e.g. PBL height or surface heating fluxes and wind speeds that may also provide prior information about the ALH.

**Author's response:** Yes, these analyses have been conducted as a part of TROPOMI's as well as Sentinel-4 and Sentinel-5 ALH algorithm development activities, and published their respective ATBDs (some of which are public and the others not) and in Nanda et al. (2018a). In theory, varying SSA makes sense, especially over land where its uncertainty is the largest source of error in the retrieved ALH (Nanda et al., 2018a). However, in practice it leads to reduced convergence rates. Because it is not well understood, and also because the goal of the paper is not in fact to introduce new variables into the state vector and instead to try and replicate the existing ALH retrieval setup as much as possible, a discussion on incorporating SSA into the feature vector has been excluded from this paper.

The meteorological parameters mentioned by the reviewer are not incorporated into the current ALH algorithm's forward model, simply because their exclusion is not as important a source of error as the biases in the retrieved product from model errors in the surface reflectance as well as the interaction of photons reflected from the aerosols and the surface.

**Changes to the manuscript:** The following sentence is amended to the end of the paragraph.

'In general there is a greater scope to add detailed information in Disamar. However, Disamar has historically incorporated many simplifications in order to reduce computational time. The current NN model is developed with the aim to mimic Disamar as much as possible, without including additional state vector elements into the retrieval, such as chlorophyll fluorescence, aerosol optical properties, cloud properties, and so on.'

**Reviewer comment (specific 7):** Page 8 Section 3.2 First Paragraph From my reading of this, the profiles are generated randomly after selecting a random set of tropomi solar-viewing geometry combinations. Naively, I would expect that the most representative way to create the training set would be to select the profiles from the ERA reanalysis corresponding to the randomly selected orbit geometries - the model probably doesn't need to reproduce a spectrum of Saharan dust for a typical Antarctic viewing geometry. Perhaps there is a heuristic argument that the way you are doing it could more reliably span the entire set of profile/viewing combinations, but if this is so you should state it in the manuscript. Ideally you would want to test the performance by comparing multiple approaches of generating the training data, though I am not sure of your computational resources

**Author's response:** Generating training data can take weeks to months with the available resources at the KNMI. The authors of the paper agree with your statement, and will add the following to the manuscript.

**Changes to the manuscript:** Added the following to the end of the first paragraph of section 3.2.

'This training data generation strategy spans the entire set of TROPOMI solar and viewing angles as well as meteorological parameters.'

**Reviewer comment (specific 8):** Page 9, Line 1: Finding the most optimal neural network configuration requires...

Using a single set for validation leads to questions about robustness in the validation. Typically cross validation e.g. k-fold methods are used to derive more accurate estimates of model performance. Have you looked into how robust the single training set is?

**Author's response:** With regards to the reviewer's comment on whether the paper

utilises neural networks trained and validated with a 'k-fold' cross-validation approach, the answer is yes. The training dataset was first shuffled and then split into a basic train-test split, which is equivalent to a 2-fold cross-validation approach. The manuscript does not mention this, and will do so in the amendment discussed in the following.

**Changes to the manuscript:** The following is added to the first paragraph in Page 9 of the manuscript.

'Finding the most optimal neural network configuration requires testing the trained neural network model. To that extent, the training data set was split into a training-testing split, where the model was trained on a majority of the training data set and tested on the remaining minority. Once trained, the model was tested again on a test data set with 100,000 scenes outside of the training data set.'

**Reviewer comment (specific 9):** Page 9, Line 6: The sigmoid function is chosen for activation... Can you show evidence for the sigmoid function outperforming the other forms?

**Reviewer comment (specific 10):** Page 9, Line 8: For each of the neural network models. . .

It would be useful showing a comparison of performance for the different models tested - I don't have an intuitive sense from the description of how different the actual results were for the different layer/node combinations, or how robust one instance of a 25000 iteration training is. For instance if I retrained the two layer model with 100+100 nodes again with a different initialization, would it still be the optimum configuration?

**Author's response:** When it comes to deciding which neural network model configuration is the most optimal depends on many factors, one of which is simply the neural network model architecture. Sometimes, adding an extra layer can result in an improvement of the neural network model, i.e. the mean squared error, summed for all

wavelengths into a single number, between the predicted and the true output spectra is low. But if this improvement is a very small number, a simpler architecture is a better alternative as it takes less time to train and compute outputs.

This is the basis of deciding whether or not a certain configuration outperforms the other. The amendment to the manuscript will include a set of plots (three plots into a single figure) and a paragraph that explain the rationale of choosing the sigmoid function, the two-layered and 100 nodes per layer choices.

If the most optimal neural network model was retrained with a different, randomly chosen initialisation of weights, the neural network model would still remain as the most optimum configuration.

Finally, there is a typographical error in this paragraph. The number of iterations for testing the configurations are 250,000 and not 25,000 (which is too few training iterations to make any concrete statement for this specific case).

**Changes to the manuscript:** A new figure, as per the recommendation of the reviewer is added. The figure numbers of the subsequent figures are automatically adjusted. The following caption is used for the figure:

'Summed loss as a function of training step for different neural network model configurations. **(a)** The neural network models have 50 nodes per each layer with a sigmoid activation function. **(b)** The neural network models have two hidden layers with each node activated by the sigmoid function. **(c)** The neural network models have two hidden layers with a 100 nodes for each layer.'

The last line of the 3rd paragraph is removed and the 4th paragraph in section 3.2 is amended as follows

'In order to test the most optimal number of layers, the most optimal number of nodes per each layer and the activation function, several neural network configurations were trained for 250,000 iterations and their summed losses (defined as $\Delta \times n_\lambda$) were compared to find out which was the best configuration. To begin, with 50 nodes per each hidden layer, three neural networks for each of the three models were trained — one-layered, two-layered and three-layered. The neural network models performed best with at least two hidden layers (Figure 2a). For all three models, their two-layered versions show a similar summed loss to their three-layered alternatives, with the summed loss for the two-layered $NN_{disamar}(K_\tau)$ showing more stability with training epoch. Because of this, a simpler two-layered architecture is chosen for all three models. Continuing on, three other architectures for each of the three models were chosen with 50, 100, and 200 nodes for each of the two hidden layers. The results that with more training steps, the choice of 100 nodes for each of the two layers has a compromise between summed training loss and simplicity (Figure 2b), especially for $NN_{disamar}(K_\tau)$. Finally, going ahead with a two-layered and 100 nodes for each layer configuration, three activation functions namely the sigmoid function, the hyperbolic tangent function (tanh) and the rectified linear unit (relu) function were tested for each of the neural network models (Figure 2c). In this case, while all functions converge to similar summed loss values by 250,000 iterations, the sigmoid function has a good compromise between training loss and stability. Figure 3 gives a graphic representation of the neural network model.'

**Reviewer comment (specific 11):** Page 11, Line 12: Although the retrieval algorithms have good agreements...

For the low aerosol loading scenes, what happens when you include both surface albedo values in the NN model?

**Author's response:** This is not known, as the trained neural network models do not include both surface albedo values, and they are also not stored in the training data set.

[Figure]

**Reviewer comment (Minor Corrections):** Equation 2: Bold the x in the forward model to be consistent with notation for vectors.

Page 7, Line 17: Change "automatic differentiation which is a powerful algorithm that computes" to "automatic differentiation which computes"

**Author's response:** Agreed.

**Changes to the manuscript:** Amended the manuscript as per the reviewer's comment.

[Figure]

**Fig. 1.**

---

## Editor Decision (ED1)

General comments: This paper presents the application of the neural network to on-line radiative transfer calculation for accelerating the operational aerosol height retrievals from TROPOMI measurements. I agree with other two reviewers; the results are very significant and interesting about the speed up of three orders of magnitude without insignificant change of the retrieval accuracy. However, I think that several parts of this paper should be revised before publication.

Specific comments:

Page 2, line 12-15: With TROPOMI ~ 50,000 pixels per orbit in many cases.

- I am not clear for this sentence, does it mean that ~ 6 % of TROPOOMI pixels are typically identified as aerosol contaminated pixels based on UV aerosol index (> 0) for retrieving aerosol heights ?

- I would like to suggest "operational retrievals are time restricted~ for retrieving aerosol layer height" to be revised like "The operational computation capability is much restricted for TROPOOMI recording approximately 1.4 million pixels within a single orbit where 50, 000 pixels are typically identify as aerosol contaminated pixels for retrieving aerosol layer height."

Page 2 Line 17: while ➔ whereas

Page 3 Line 23: scaled by ➔ constrained with

Page 3 Line 26-28: This cost function is also constrained with a priori knowledge of the state vector x. The final retrieval product of zaer and $\tau$

Page 4 line 1: "The forward model is employed to simulate the measured reflectance spectrum with model parameter $x$ as following" or delete "modeled"

Page 4 line 4: delete ")" from wavelength $\lambda$)

Where I and Eo represent the Earth radiance and solar irradiance, respectively, with the cosine of the solar zenith angle $(\theta_0)$ $u_o$

Page 4 line 5: employed to update the state vector as following

Page 4 line 7: where  Ki is the matrix of derivatives (Jacobian) of the reflectance with respect to state vector parameters at the current iteration i.

Page 4 line 9: An iterative estimate is convergent to a solution if the relative changes in the state vector is less than ~.

Page 4 Line 10-11: The retrieval is decided to be failed if ~ /their respective boundary conditions by OE ➔ the respective boundary conditions

Page 4 Line 15: The forward model iteratively simulates TOA radiance spectra until the convergence of $x^2$ (Equation 1).

Page 4 Line 16: To define TOA reflectance, convolved high resolution reference solar spectrum onto the instrument's slit function is used instead of measured solar irradiance?

Page 5 Line 4 : While ➔ In spite that

Page 5 Line 5: (Sanders and de Haan, 2016) ➔ Sanders and de Haan, 2016

or preliminary experiments have ~~ (Sanders and de Haan, 2016).

Page 5 Line 5-6: the impact of ignoring RRS on the retrieval of aerosol layer height using the oxygen A-band are much smaller than that of other retrieval errors such as ??.

Page 5 Line 6-7: Due to this~ cross sections ➔ Therefore, RRS has been historically not simulated in the forward model of the KNMI aerosol layer height retrieval algorithm.

Page 5 Line 8-9: The retrieval of Zaer in the presence of clouds is still challenging (reference) and thereby is performed only for cloud-free cases masked when cloud fraction is less than 0.2. Compared to totally cloud-free scene, the retrieval errors of Zaer are considerably problematic when the measured scene is masked as clear-sky in the presence of optically thin cirrus (reference).

Page 5 Line 10 : "Instead, TROPOMI incorporates information from the VIIRS instrument to detect the

Presence of clouds in the measured scene, which are further on mentioned in the output product flags

Instead"

 - This author did not describe how to identify cloudy scenes up to here in previous algorithm or instrument instead current TROPOMI algorithm.

-  "TROPOMI" ➔ TROPOMI Zaer algorithm or TROPOMI cloud algorithm?

-  "which are further on mentioned in the output product flags" ➔ What is output product here? Aerosol height or cloud product? Anyway output product flag contains information on how to identify a pixel as cloud using cloud product between TROPOOMI or VIIRS? Please specify more.

Page 5 Line 11: Please provide details on how to decide the fraction of the pixel containing aerosols.

Page 5 Line 15-16 : The aerosol scattering phase function ~ significantly more computations ➔ A Henyey-Greenstein model (Henyey and Greenstein, 1941) is used to parametrize the aerosol scattering phase function, which is one of the widely used approximations.

Page 5 Line 17: what is the fixed aerosol optical properties taken from AERONET data?

I am not clear about "the consequences of fixing them" based on the following sentence "GOME-2 spectra to show that the algorithm is robust against these model assumptions"

Page 5 Line 26: Actually, does NN-based algorithm include any complexity of the model thanks to the speed up? If not, please revise "In constrat ~ endeavor" to "The speed up of forward model simulation encourages increasing the complexity of simulation assumption ~~" and move to section 5.

Page 6 Line 6: modeled measured reflectance ➔ modeled reflectance

Page 6 Line 8: are derived from ➔ are taken from|, which provide ➔, including.

What is about meteorological input at surface level?

Page 6 Line 9: The various ~ parameters == > The various geophysical parameters

Page 6 Line 11: requires ➔ takes?

Page 7 Line 4: Kingma and Ba (2014)) ➔ Kingma and Ba (2014)

Page 9 Line 34: Because of this == > Therefore

Page 12 : (a) I am not clear why "augmented" is used as an adjective for the neural network. It looks better just to indicate it as "the neural network". (b) in this section, most of Figures are not directly

introduced such as ~~~(Table 4). Please try to introduce a Figure directly and then give relative analysis.

Page 12 Line 2: Figure 8a ➜ Figure 8b | insert Figure a before ", which" at line 1. Please use Figure 8b more in this analysis.

Page 12 line 2-3: absorbing aerosol index (AAI)

Page 12 line 4: Pixels that were cloud contaminated ➜ cloud-contaminated pixels

Page 12 line 4: What is the processing chain?

Page 12 line 5-6:

-Scientific comment: the FRESCO-based cloud fraction is positively biased for all typed aerosols or just for biomass burning aerosols. Please clarify.

-Editing comment: "However, the cloud-free biomass burning aerosol pixels could be screened out as the high cloud fraction of greater than the threshold is likely to be retrieved."

Page 12 line 7: ", as the surface ~ in these regions" ➜ where the surface albedo retrieval is likely to be wrong.

Page 12, line 3-7: This part should be described in detail in section 2: the TROPOMI aerosol layer height retrieval algorithm.

Page 12, line 8-10: Figure 9 compares the retrieved Zaer over the plume using the line-by-line and neural network based forward models, respectively. The number of the converged retrievals is 7418 for the line-by-line algorithm, but 7370 for the neural network algorithm.

Page 12, line 11: this analysis is contradictory, please revise and give more interprets on Fig 8.c; for example, where/why the positive/negative biases are dominant.

Page 12, line13-15: please revise this sentence, it is very hard to see what is the subject for "indicate" after respectively.

Page 12, line 18: due to over-estimation ➜ caused by over-estimation of ?? by.

Page 12, line 19: a consistent bias of 60 meters with a standard deviation of 30 meters.

Page 13, line 2: the aerosol layer height algorithm among L2 algorithms is unique for implementing on-line RT?

Page 13, line 4: Disamar just calculate radiance?

Page 13, line 22: We evaluate the Zaer retrieved from TROPOMI measurements over Southern California on 12 December 2017 when the fire plume extensively floats from land to ocean over a dry and almost cloudless scene.

Table 4 caption: Statistics of difference in retrieved zaer between Disamar and NN from figure 9c.

Figure 2, Figure 3, Figure 4: characters looks vague.

---

## Author Response (AR2)

Author's response to Associate editor's comments for the manuscript "*A neural network radiative transfer model approach applied to TROPOMI's aerosol height algorithm*" (amt-2019-143).

**Reviewer comment (general):** This paper presents the application of the neural network to on-line radiative transfer calculation for accelerating the operational aerosol height retrievals from TROPOMI measurements. I agree with other two reviewers; the results are very significant and interesting about the speed up of three orders of magnitude without insignificant change of the retrieval accuracy. However, I think that several parts of this paper should be revised before publication.

**Author's response:** Thank you for taking the time to review this manuscript.

**Reviewer comment (specific 1):** Page 2, line 12-15: With TROPOMI 50,000 pixels per orbit in many cases.

- I am not clear for this sentence, does it mean that 6% of TROPOMI pixels are typically identified as aerosol contaminated pixels based on UV aerosol index ($> 0$) for retrieving aerosol heights ?

- I would like to suggest "operational retrievals are time restricted for retrieving aerosol layer height" to be revised like "The operational computation capability is much restricted for TROPOMI recording approximately 1.4 million pixels within a single orbit where 50,000 pixels are typically identify as aerosol contaminated pixels for retrieving aerosol layer height."

Page 2 Line 17: while → whereas

**Author's response:**

- Yes, it does mean that approximately 6% of all TROPOMI pixels over an area as large as Europe will contain UVAI values above 0.0.

- We accept the suggestion.

Finally the comment **Page 2 Line 17: while → whereas** is rejected as that is incorrect grammar and changes the sentence structure.

The manuscript will clarify the sentence and include suggestions made by the Associate Editor.

**Changes to the manuscript:**

The paragraph now reads (with bold characters representing the change):

As near-real time processors need to consistently go through large volumes of data recorded by the satellite for the mission lifetime, **the operational computation capability is much restricted for TROPOMI recording approximately 1.4 million pixels within a single orbit where, on average, 50,000 pixels are typically identified as aerosol contaminated pixels (with a UVAI value greater than 0.0) for retrieving aerosol layer height. This places a steep requirement** on the computational infrastructure to process all possible pixels from a single orbit. The online radiative transfer model severely limits the ALH data product, processing only a small fraction of the total possible pixels within a single orbit **while** compromising the timeliness of the data delivery.

**Reviewer comment (technical comments) and response:**

Page 3 Line 23: scaled by → constrained with

**Response: Accepted.**

Page 3 Line 26-28: This cost function is also constrained with a priori knowledge of the state vector x. The final retrieval product of zaer and $\tau$

**Response: Accepted. The paragraph is amended as follows:**

Minimising this cost function for a particular $z_{\text{aer}}$ and $\tau$ (the elements of the state vector $x$ to be retrieved and fitted) gives us the final retrieval product. **This definition of the cost function is unique to OE, as it is constrained with a priori knowledge of the state vector $x$ (represented by $x_a$) and the a priori error covariance matrix $\mathbf{S_a}$.**

Page 4 line 1: "The forward model is employed to simulate the measured reflectance spectrum with model parameter $x$ as following" or delete "modeled"

**Response: Accepted.**

Page 4 line 4: delete ")" from wavelength $\lambda$)

**Response: Accepted.**

Where I and $E_o$ represent the Earth radiance and solar irradiance, respectively, with the cosine of the solar zenith angle $(\theta_0)$ $\mu_0$

**Response: Accepted.**

Page 4 line 5: employed to update the state vector as following

**Response: Accepted.**

Page 4 line 7: where  Ki is the matrix of derivatives (Jacobian) of the reflectance with respect to state vector parameters at the current iteration i.

**Response: Accepted.**

Page 4 line 9: An iterative estimate is convergent to a solution if the relative changes in the state vector is less than .

**Response: Accepted.**

Page 4 Line 10-11: The retrieval is decided to be failed if   /their respective boundary conditions by OE → the respective boundary conditions

**Response: Accepted.**

Page 4 Line 15: The forward model iteratively simulates TOA radiance spectra until the convergence of $\chi^2$ (Equation 1).

**Response: Accepted with minor changes: Optimal estimation iteratively simulates TOA radiance spectra until the convergence of $\chi^2$ (Equation 1).**

Page 4 Line 16: To define TOA reflectance, convolved high resolution reference solar spectrum onto the instrument's slit function is used instead of measured solar irradiance?

**Response: The paragraph is not clear on how the forward model calculations are done. The following is the amendment to the manuscript:**

**For this, disamar computes reflectances at a high resolution wavelength grid. The computed high resolution reflectances are combined with a reference solar spectrum derived from Chance and Kurucz (2010) to obtain a high resolution Earth radiance. The high resolution Earth radiance and the solar spectrum are convolved with the instrument spectral response function to obtain Earth radiance and solar irradiance spectrum in the instrument's wavelength grid, before finally computing the reflectance spectrum in the instrument grid using Equation 2.**

Page 5 Line 4 : While → In spite that

**Response: not accepted. 'In spite that' is incorrect grammar. The sentence is unamended.**

Page 5 Line 5: (Sanders and de Haan, 2016) → Sanders and de Haan, 2016 or preliminary experiments have (Sanders and de Haan, 2016).

**Response: accepted.**

Page 5 Line 5-6: the impact of ignoring RRS on the retrieval of aerosol layer height using the oxygen A-band are

much smaller than that of other retrieval errors such as ??.

**Response: accepted. Amended as follows:**

ignoring RRS of the oxygen A-band in the forward model are significantly smaller than the effect of other model errors **such as errors due to incorrect surface albedo**.

Page 5 Line 6-7: Due to this  cross sections → Therefore, RRS has been historically not simulated in the forward model of the KNMI aerosol layer height retrieval algorithm.

**Response: accepted.**

Page 5 Line 8-9: The retrieval of Zaer in the presence of clouds is still challenging (reference) and thereby is performed only for cloud-free cases masked when cloud fraction is less than 0.2. Compared to totally cloud-free scene, the retrieval errors of Zaer are considerably problematic when the measured scene is masked as clear-sky in the presence of optically thin cirrus (reference).

**Response: accepted with minor changes to suggestion:**

The atmosphere is assumed cloud-free, which is a required simplification as the retrieval **of $z_{aer}$ in the presence of clouds is still challenging (Sanders et al., 2015) and thereby is performed only for pixels which are unlikely to contain clouds. Compared to totally cloud-free scenes, errors in retrieved $z_{aer}$ are large for cloud-free scenes containing undetected optically thin cirrus clouds (Sanders et al., 2015).**

**Reviewer comment (specific comments):**

Page 5 Line 10 : "Instead, TROPOMI incorporates information from the VIIRS instrument to detect the Presence of clouds in the measured scene, which are further on mentioned in the output product flags"

- This author did not describe how to identify cloudy scenes up to here in previous algorithm or instrument instead current TROPOMI algorithm.

- "TROPOMI" → TROPOMI Zaer algorithm or TROPOMI cloud algorithm?

- "which are further on mentioned in the output product flags" → What is output product here?

- Aerosol height or cloud product? Anyway output product flag contains information on how to identify a pixel as cloud using cloud product between TROPOMI or VIIRS? Please specify more.

**Author's response:** Cloud detection is not a topic of this paper and hence is not convered in this paper. However, in order to reduce confusion, cloud flagging is further clarified.

**Changes to the manuscript:** The paragraph has been removed from section 2.2 and added to 2.3. It now reads:

**TROPOMI incorporates information from the VIIRS instrument to detect the presence of cirrus clouds in the measured scene (using a cirrus reflectance threshold of 0.01). This information is further combined with cloud fraction retrievals by the TROPOMI FRESCO algorithm (maximum cloud fraction of 0.6), and the difference between the scene albedo in the database in the UV band and the apparent scene albedo at the same wavelength calculated using a lookup table (if the difference is larger than 0.2, it suggests cloud contamination). A combination of these different cloud detection strategies results in the cloud_warning flag in the level-2 TROPOMI ALH product.**

**Reviewer comment (specific comments):**

Page 5 Line 11: Please provide details on how to decide the fraction of the pixel containing aerosols.

**Author's response:** Aerosol fraction is not a retrieval parameter, and so there are no details available to decide the fraction of the pixel containing aerosols. The pixel is simply assumed to entirely contain aerosols.

**Changes to the manuscript:** There are no changes made to the manuscript for this comment.

**Reviewer comment (technical comment):**

Page 5 Line 15-16 : The aerosol scattering phase function   significantly more computations → A Henyey-Greenstein model (Henyey and Greenstein, 1941) is used to parameterize the aerosol scattering phase function, which is one of the widely used approximations.

**Author's response:** Accepted. The sentence has been amended as suggested by the reviewer:

**A Henyey-Greenstein model (Henyey and Greenstein, 1941) with an asymmetry parameter value of 0.7 is used to parameterize the aerosol scattering phase function, which is one of the widely used approximations.**

**Reviewer comment (specific comment):**

Page 5 Line 17: what is the fixed aerosol optical properties taken from AERONET data?

I am not clear about "the consequences of fixing them" based on the following sentence "GOME-2 spectra to show that the algorithm is robust against these model assumptions"

**Author's response:** The fixed optical properties mentioned in this sentence are the single scattering albedo and the Henyey Greenstein phase function asymmetry parameter.

**Changes to the manuscript:** The sentence now reads:

**These fixed aerosol optical properties have been derived from AERONET data and tested by Sanders et al. (2015), who retrieved $z_{aer}$ from GOME-2 spectra to show that the retrieval algorithm is robust to fixing aerosol model parameters such as the single scattering albedo and the Henyey-Greenstein phase function asymmetry parameter.**

**Reviewer comment (specific comment):**

Page 5 Line 26: Actually, does NN-based algorithm include any complexity of the model thanks to the speed up? If not, please revise "In constrat   endeavor" to "The speed up of forward model simulation encourages increasing the complexity of simulation assumption    " and move to section 5.

**Author's response:** Accepted.

**Changes to the manuscript:** The following sentences were REMOVED:

*In contrast, a neural network model is significantly faster. While the speed of the neural network model encourages increasing the complexity of the model, for a comparative study the neural network models are trained to replicate, as best as possible, the line-by-line version. Once this is achieved, the improvement of the algorithm will be an iterative endeavour.*

The paragraph now reads:

**These simplifications in the Disamar forward model are a necessity for the line-by-line aerosol layer height algorithm, owing to its slow computational speed. The speed up of forward model simulation encourages increasing the complexity of simulation assumption.**

**Reviewer comment (technical comments):**

Page 6 Line 6: modeled measured reflectance → modeled reflectance

**Response: accepted.**

Page 6 Line 8: are derived from → are taken from, which provide →, including.

What is about meteorological input at surface level?

**Response: accepted. The meteorological input at surface level is also derived from ECMWF. The**

sentence is amended as follows:

Meteorological parameters are taken from ECMWF (European Centre for Medium-range Weather Forecast), including the temperature-pressure profile at 91 atmospheric levels (of which the surface is a part).

Page 6 Line 9: The various   parameters → The various geophysical parameters

**Response: accepted.**

Page 6 Line 11: requires → takes?

**Response: accepted.**

Page 7 Line 4: Kingma and Ba (2014)) → Kingma and Ba (2014)

**Response: accepted.**

Page 9 Line 34: Because of this → Therefore

**Response: accepted.**

**Reviewer comment (specific comment):**

Page 12 :

  (a) I am not clear why "augmented" is used as an adjective for the neural network. It looks better just to indicate it as "the neural network".

  (b) in this section, most of Figures are not directly introduced such as   (Figure 9a),   (Table 4). Please try to introduce a Figure directly and then give relative analysis.

**Author's response:**

  (a) Accepted. The word "augmented" is removed.

  (b) Accepted. The following changes have been made to the manuscript:

  **Section 3.2: Figure 1 plots the distribution of the input parameters necessary for training the neural network. The neural network model accepts solar azimuth and viewing azimuth angles separately, however they are plotted together as relative azimuth angle in Figure 1 to save space.**

  **Section 3.2: Figure 2 plots the summed losses as a function of training iteration for different configurations.**

  **Section 3.2: Figure 4 plots the performance of each of the neural networks trained on the testing data set**

  **Section 4.1: Figure 5 compares the retrieved $z_{\mathrm{aer}}$ from line-by-line and neural network approaches for each of the synthetic experiments. A histogram of these differences in plotted in Figure 6.**

  **Section 4.1: Out of the 8000 scenes within the synthetic experiment, NN retrieved aerosol layer heights for 546 scenes where Disamar did not. Contrariwise, 586 scenes converged for Disamar and not for NN. A comparison of the biases from these odd retrieval results is plotted in Figure 7, which indicates that retrievals from NN in cases where Disamar fails are realistic as the distribution of the biases is very similar to those cases when Disamar succeeds and NN does not (Figure 7)**

  **Section 4.2: A MODIS Terra image of the plume and the retrieved absorbing aerosol index from TROPOMI is plotted in Figure 8.**

  **Section 4.2: Figure 9 compares the retrieved $z_{\mathrm{aer}}$ over the plume using the line-by-line and neural network based forward models, respectively.**

**Section 4.2: Figure 10 provides plots for further comparison between the two retrieval techniques.**

**Reviewer comment (technical comments):**

Page 12 Line 2: Figure 8a → Figure 8b — insert Figure a before ", which" at line 1. Please use Figure 8b more in this analysis.

**Response: accepted.**

Page 12 line 2-3: absorbing aerosol index (AAI)

**Response: accepted.**

Page 12 line 4: Pixels that were cloud contaminated → cloud-contaminated pixels

**Response: accepted.**

Page 12 line 4: What is the processing chain?

**Author's response: processing chain here refers to the pool of pixels to be processed. To make it more clear, the sentence has been amended in the following manner:**

Cloud-contaminated pixels were removed from the data selected for processing using the FRESCO cloud mask product from TROPOMI (maximum cloud fraction of 0.2)

**Reviewer comment (specific comments):**

Page 12 line 5-6:

- Scientific comment: the FRESCO-based cloud fraction is positively biased for all typed aerosols or just for biomass burning aerosols. Please clarify.

- Editing comment: "However, the cloud-free biomass burning aerosol pixels could be screened out as the high cloud fraction of greater than the threshold is likely to be retrieved."

**Author's response:**

- In general, FRESCO-based cloud fraction retrievals will be positively biased over a cloud-free scene containing aerosols. The manuscript was amended as:

  **... removed as the cloud fraction values for these pixels were higher than the threshold. This is because FRESCO-based cloud fraction values over cloud-free scenes containing aerosols (biomass burning aerosols in this case) are generally expected to be positively biased. The retrieval algorithms did ...**

- This is not correct as we have observed accurate aerosol layer height retrievals (when compared to co-located lidar profiles) for pixels with FRESCO cloud fraction threshold values greater than 0.6 as well. This editing comment is not implemented into the manuscript.

**Reviewer comment (technical comments):**

Page 12 line 7: ", as the surface   in these regions" → where the surface albedo retrieval is likely to be wrong.

**Response: accepted.**

Page 12, line 3-7: This part should be described in detail in section 2: the TROPOMI aerosol layer height retrieval algorithm.

Response: accepted. The following paragraph has been added to Section 2.3:

TROPOMI incorporates information from the VIIRS instrument to detect the presence of cirrus clouds in the measured scene (using a cirrus reflectance threshold of 0.01). This information is further combined with cloud fraction retrievals by the TROPOMI FRESCO algorithm (maximum cloud fraction of 0.6), and the difference between the scene albedo in the database in the UV band and the apparent scene albedo at the same wavelength calculated using a lookup table (if the difference is larger than 0.2, it suggests cloud contamination). A combination of these different cloud detection strategies results in the cloud_warning flag in the level-2 TROPOMI ALH product.

Page 12, line 8-10: Figure 9 compares the retrieved Zaer over the plume using the line-by-line and neural network based forward models, respectively. The number of the converged retrievals is 7418 for the line-by-line algorithm, but 7370 for the neural network algorithm.

Response: accepted.

Page 12, line 11: this analysis is contradictory, please revise and give more interprets on Fig 8.c; for example, where/why the positive/negative biases are dominant.

Response: accepted. The paragraph now reads:

The differences between $z_{\mathrm{aer}}$ (disamar) and $z_{\mathrm{aer}}$ (NN) go up to as much as 0.5 km (Figure 9c). A majority of the negative differences are for the part of the plume extending from the coast between 47°N and 40°N. Figure 10 provides plots for further comparison between the two retrieval techniques.

The paragraph goes on further to answer the conditions where the negative/positive biases are dominant by discussing Figure 10. Figure 9c alone is insufficient to answer this question.

Page 12, line13-15: please revise this sentence, it is very hard to see what is the subject for "indicate" after respectively.

Response: agreed. The paragraph now reads:

... retrieved aerosol layer heights which were (on average) less than 50.0 meters apart from the same by the line-by-line counterpart (Figure 10b). The standard deviation of the differences are approximately 160 meters, which indicates the presence of outliers. However, a majority of the differences in the two retrievals are less than 100 meters; this is indicated by the $15^{\mathrm{th}}$ and the $85^{\mathrm{th}}$ percentile of these differences of -115.0 meters and 40.0 meters respectively. Although ...

Page 12, line 18: due to over-estimation $\rightarrow$ caused by over-estimation of ?? by.

Response: accepted. The sentence now reads:

Most of these biases were caused by an over-estimation of the retrieved aerosol layer height using the neural network algorithm, in comparison to the same from disamar.

Page 12, line 19: a consistent bias of 60 meters with a standard deviation of 30 meters.

Response: accepted.

Page 13, line 2: the aerosol layer height algorithm among L2 algorithms is unique for implementing online RT?

Response: no. There are several other algorithms that use online radiative transfer calculations. The statement does not say that aerosol layer height is unique in its use of online RT calculations among every other level 2 algorithm, however it is clarified as follows:

Of the algorithms that currently retrieve TROPOMI's suite of level-2 products, the aerosol layer height processor is an example of one that requires online radiative transfer calculations.

Page 13, line 4: Disamar just calculate radiance?

Response: no. Disamar calculates several outputs, however the sun-normalised radiances are the important one for aerosol layer height (as well as the derivatives). The line is clarified as follows:

These online calculations have traditionally been tackled with KNMI's radiative transfer code disamar,

**which calculates (among other parameters) sun-normalised radiances in the oxygen A-band.**

Page 13, line 22: We evaluate the Zaer retrieved from TROPOMI measurements over Southern California on 12 December 2017 when the fire plume extensively floats from land to ocean over a dry and almost cloudless scene.

**Response: accepted with minor changes to the reviewer's suggestion. The paragraph now reads:**

**We evaluate aerosol layer heights retrieved from TROPOMI measurements over Southern California on 12 December, 2017, when the fire plume extensively floats from land to ocean over a dry and almost cloudless scene.**

Table 4 caption: Statistics of difference in retrieved zaer between Disamar and NN from figure 9c.

**Response: accepted.**

Figure 2, Figure 3, Figure 4: characters looks vague.

**Response: accepted. The font sizes in the plots have now been increased.**

[revised manuscript text omitted]

---

## Author Response (AR3)

Author's response to Associate editor's comments for the manuscript "*A neural network radiative transfer model approach applied to TROPOMI's aerosol height algorithm*" (amt-2019-143).

**Reviewer comment (Minor Corrections):** Page 2, line 26 : O2-O2 O2-O2 (last '2' not in subscript)

**Author's response:** corrected.

**Changes to the manuscript:** changed the notation in accordance to the associate editor's recommendation.

**Reviewer comment (Minor Corrections):** Page2, line 28: MODIS aerosol optical depth at 550 nm product ==> MODIS aerosol optical depth at 550 nm.

**Author's response:** corrected.

**Changes to the manuscript:** removed the word 'product'.

**Reviewer comment (Minor Corrections):** Page 2, Line 30: 1) They demonstrated: here "they" should be matched to "their neural network models", please amend. 2) OMI is here indicated such as "the Ozone Monitoring Instrument (OMI) on board the NASA Aura mission, it looks like that OMI is first mentioned here, but already mentioned many times such as line 26 and 29 in the same page.

**Author's response:** corrected.

**Changes to the manuscript:** amended as per recommendation of the associate editor.

**Reviewer comment (Minor Corrections):** Page 3. Some references are in wrong foramt, e.g. without year of publication: e.g., lines 3, 4, 6 (e.g., Loyola et al. instead of Loyola et al. 2018).

**Author's response:** accepted.

**Changes to the manuscript:** the references are now corrected.

**Reviewer comment (Minor Corrections):** Page 4, line 14: n iterative estimate The nth iterative estimate?

**Author's response:** corrected.

**Changes to the manuscript:** changed as per associate editor's recommendation.

**Reviewer comment (Minor Corrections):** In 2.1, any explanation for Sa?

**Author's response:** the explanation for Sa is already in the text: *This definition of the cost function is unique to OE, as it is constrained with a priori knowledge of the state vector x (represented by $x_a$) and the a priori error covariance matrix $\mathbf{S_a}$.*

**Changes to the manuscript:** no changes made to the manuscript.

**Reviewer comment (Minor Corrections):** Section 2.2, For simulating the measured reflectance, the high resolution reflectance () calculated from NN is directly convolved with instrument slit function ( ), but you convert reflectance to radiance (at high resolution and then perform the convolution for and , separately for finally getting I/F, which is indicated by "Io correction". I am curious that there is significant impact on O2 absorption band with and without Io correction. I think that it could be very useful for readers to describe why you perform Io correction for your convolution process.

**Author's response:** It is important to note that the neural network aerosol layer height retrieval algorithm does not incorporate the Io correction you mention. This correction is only for disamar. The difference of excluding

[Figure]

Figure 1: The difference (reported as a percentage) between the two convolution processes, one with Io correction and the other without.

and including the Io correction at most 4% (w.r.t convolving while including the Io correction). This has been determined by the following: convolve a high resolution reflectance spectrum using the "Io correction" and do the same experiment without "Io correction". The following figure reports the difference in percentage) between the two w.r.t the method using "Io correction":

While this difference may be considered noteworthy, the simulation experiments in Figure 5 of the paper compare retrieved aerosol layer heights from both methods - disamar with the correction and neural networks which does not use this correction method. The figure shows that the retrieved ALH(s) are very similar.

The manuscript will be updated to reflect the above, and the figure describing these differences will not be included.

**Changes to the manuscript:** The following text is added to Section 2.2, paragraph 1:

*It is important to note that the steps of including the reference solar spectrum to compute reflectances in the instrument's wavelength grid are not undertaken by the neural network algorithm. The neural network aerosol layer height retrieval algorithm directly convolves the reflectance. The difference between including an excluding a reference spectrum in the convolution process results in differences in the order of 4% to 5% around 762 nm and 766 nm. Further on in this paper, a direct comparison between disamar retrievals of aerosol layer height and retrievals with the neural network algorithm is provided.*

**Reviewer comment (Minor Corrections):** Page 5, line 2: aerosols, the surface and molecular species aerosols, surface, and molecular species ?

**Author's response:** adjusted.

**Changes to the manuscript:** adjusted as per associate editor's recommendations.

**Reviewer comment (Minor Corrections):** Page6, line 17: the ozone monitoring instrument (OMI) OMI

**Author's response:** accepted.

**Changes to the manuscript:** removed 'the ozone monitoring instrument'

**Reviewer comment (Minor Corrections):** Page 8, line 33: a fixed g value the symbol g is not specified, it could be good to denote g in page 5, line 29 in section of 2.2. In addition, the symbol omega for single scattering albedo could be first denoted in section of 2.2.

**Author's response:** accepted.

**Changes to the manuscript:** changed as per associate editor's recommendation.

**Reviewer comment (Minor Corrections):** Page 11, line 33: A histogram of these differences in plotted in Figure 6 is plotted

**Author's response:** agreed.

**Changes to the manuscript:** corrected as per associate editor's recommendation.

**Reviewer comment (Minor Corrections):** Page 13, line 1-2: "Absorbing aerosol index (AAI) should be placed in the first line rather than the second line.

**Author's response:** the manuscript has been adjusted overall to avoid redundancy.

**Changes to the manuscript:** The abbreviation of AAI is introduced in Table 1.

**Reviewer comment (Minor Corrections):** Page 14, line 14: each of were each of was

**Author's response:** accepted.

**Changes to the manuscript:** corrected as per recommendation.

[revised manuscript text omitted]